# A deep learning ECG model for identification and localization of occlusion myocardial infarction

Stefan Gustafsson [1], Antônio H. Ribeiro [2,3], Daniel Gedon [4,5], Petrus E. O. G. B. Abreu[6], Nicolas Pielawski[2], Gabriela M. M. Paixão[7], Marco Antonio Gutierrez [8], José Eduardo Krieger [8], Felipe Meneguitti Dias [8], Antonio Luiz P. Ribeiro [7], Daniel Lindholm [1,9], Thomas B. Schön[2] & Johan Sundström [1,10] ✉

Rapid identification and localization of an acute coronary occlusion are vital to prevent myocardial damage, yet reliance on ST-segment ECG criteria misses many acute occlusion myocardial infarctions (OMI) and triggers unnecessary acute angiographies. Here, we present a trained and validated deep learning model using 540,372 emergency ECGs paired with definitive catheterization outcomes. The model has a C-statistic of ≥0.95 for OMI and ≥0.87 for non-OMI infarctions and can localize culprit lesions in the three main coronary branches, which can guide the angiographer. Performance is similar across age, sex, and ECG hardware subgroups. Obviating dependence on ST-elevations and troponins, this model for the identification and localization of OMI has the potential to shorten the time to reperfusion of an acute coronary occlusion and save resources. Because human oversight of OMI detection on the ECG is limited, randomized clinical trials with patient-relevant outcomes are warranted.

Acute myocardial infarction is the leading cause of death globally[1]. When a patient presents with a suspected acute coronary syndrome, the goal is to decide if the patient needs to be urgently transported to a coronary intervention lab or if the angiography can wait, e.g., until office hours. If the coronary artery is acutely occluded, time equals dying myocardium. The electrocardiogram (ECG) is, therefore, of great utility because it can often be obtained already in the ambulance and could herald an acute occlusion before permanent myocardial damage entails. The current ECG paradigm mainly utilizes the ST-segment for these decisions, but ST-elevations are only present in a fraction of

acute occlusion myocardial infarctions (OMI)[2], ST-elevations can have other causes than OMI, and physicians disagree in their interpretations of ST-elevations[3,4]. This leads to lost time[5,6], lost lives[2], and costs and risks associated with unnecessary coronary angiographies.

Recent dramatic advances in ECG interpretation using machine learning offer the potential to better identify an OMI, potentially saving time, lives, and costs[7]. Machine learning models using parameterized[8] or raw[9] ECG data have shown promise for OMI classification. The ECG can also be useful for determining the localization of pathologies, and machine learning models have been shown to have the capacity to

[1]Department of Medical Sciences, Clinical Epidemiology Unit, Uppsala University, Uppsala, Sweden. [2]Department of Information Technology, Division of Systems and Control, Uppsala University, Uppsala, Sweden. [3]Science for life laboratory (SciLifeLab), Uppsala, Sweden. [4]Machine Learning in Science, University of Tübingen, Tübingen, Germany. [5]Tübingen AI Center, Tübingen, Germany. [6]Postgraduate Program in Health Sciences: Infectious Diseases and Tropical Medicine, Universidade Federal de Minas Gerais, Belo Horizonte, Brazil. [7]Department of Internal Medicine, Faculdade de Medicina, and Telehealth Center and Cardiology Service, Hospital das Clínicas, Universidade Federal de Minas Gerais, Belo Horizonte, Brazil. [8]Heart Institute, University of São Paulo Medical School, São Paulo, Brazil. [9]Department of Medicine, Norrtälje hospital (Tiohundra AB), Norrtälje, Sweden. [10]The George Institute for Global Health, University of New South Wales, Sydney, NSW, Australia. ✉e-mail: johan.sundstrom@uu.se

## A

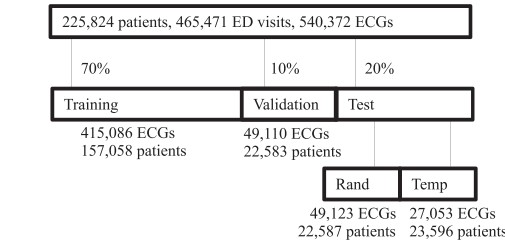

Adult (≥18y) all-comer ED patients with ECG passing quality control, taken within 1 day of ED visit, in the Karolinska ECG database, Stockholm, Sweden 2005-2016 with available registry data

227,844 patients, 474,537 ED visits, 581,273 ECGs

1,233 ED visits with MI 2005-2006 (when exact MI subtype is unavailable)
3,983 ED visits with MI without angiography within 7 days
3,618 ED visits not assignable to a single, unique ED only or CCU record with available outcome labels, or with conflicting outcome labels
508 ECGs with CABG/PCI performed before ECG recording

225,824 patients, 465,471 ED visits, 567,438 ECGs

27,066 repeated ECG recordings of same admission in testsets (one recording kept at random)

225,824 patients, 465,471 ED visits, 540,372 ECGs

Per admission:
461,386, 3,017, 1,068 control, nOMI, OMI

Per ECG:
534,510, 4,279, 1,583 control, nOMI, OMI

## B

225,824 patients, 465,471 ED visits, 540,372 ECGs

70%          10%          20%

Training     Validation    Test

415,086 ECGs    49,110 ECGs
157,058 patients  22,583 patients

Rand    Temp

49,123 ECGs   27,053 ECGs
22,587 patients  23,596 patients

**Fig. 1 | Flowchart describing the creation of the main study sample.** Panel **A** shows inclusion and exclusion criteria for the main study sample (the Swedish Emergency Department database [SwED]). There are three main quantities where N ECGs > N patient visits > N patients. Panel **B** shows data splits of the derived main study sample. The sets (training, validation, test random, test temporal) are created so that a patient can only belong to one set (to prevent any information leakage between the sets). The validation and test sets are further restricted so they use at maximum one ECG per patient visit. The training, validation and random test set includes patients drawn at random whereas the temporal test set includes patients with their first available ECG recording during the last year (2016).

localize a non-occlusive coronary stenosis, i.e., the potential to localize coronary artery narrowing in the absence of a complete occlusion[10]. Some patients have chronic coronary occlusions in addition to an ongoing acute OMI, making the choice of which coronary occlusion to intervene on difficult but critical[11]. Correctly identifying the localization of the acute occlusion would therefore be of great clinical value. We developed and validated such a machine learning model for the identification of OMI and its localization, by using more than half a million ECGs and detailed anatomical coronary angiography and intervention results in emergency department patients.

## Results and discussion
### Study population
We utilized 540,372 ECGs from 465,471 emergency department (ED) visits and 225,824 patients in the Swedish Emergency Department Database (SwED; Fig. 1)[12]. Of these ECGs, 1583 (0.3%), 4279 (0.8%), and 534,510 (98.9%) were labeled as OMI, nOMI (myocardial infarction not classified as OMI), and control (no myocardial infarction), respectively. The OMI and nOMI labels were set at time of coronary catheterization by the coronary angiographer. OMI was defined as the presence of a newly formed coronary total occlusion or very close to total occlusion (TIMI blood flow 0). The distinction between an acute and a chronic occlusion can be made with certainty during the coronary

intervention, based on the characteristics of the occlusion upon guidewire maneuvering. Of those with an OMI, 595 (37.6%), 359 (22.7%), and 629 (39.7%) had the left main or left anterior descending artery (LM/LAD), the left circumflex artery (LCX), and the right coronary artery (RCA) as the culprit vessel (the acutely blocked artery responsible for this myocardial infarction), respectively (Fig. 2A, B).

We split the SwED data on patients (not ECGs), using 70% for training, 10% for validation, and 20% for testing. The test data comprised one random set and one temporal set (including patients whose first visit occurred in the final study year), which is further described in Fig. 1. The numbers in each outcome class (defined in Supplementary Fig. 1) and data splits are shown in Supplementary Table 1, and the numbers in each sub-class over time are shown in Supplementary Figs. 5, 6. The number of samples in each sub-class increased over the years. This increase was likely due to multiple factors, e.g. including expanded ECG data coverage and changes in clinical care such as a growing proportion of non ST-elevation myocardial infarction (NSTEMI) and elderly patients with available angiography, as shown in the SWEDEHEART Annual report 2024[13].

Compared with the temporal test set, the random test set included more patients with multiple ED visits in the SwED data (over a longer time span), were on average older, and had more prevalent cardiovascular comorbidities. The clinical characteristics by OMI class are shown in Supplementary Table 6 for all data splits of SwED combined, showing that the OMI cases were on average younger than the nOMI cases, had a higher proportion of men, and had fewer known underlying cardiovascular comorbidities and use of cardiovascular drugs at the time of the ED visit. The OMI cases also have markedly elevated levels of cardiac troponin levels, an elevated risk of death within 30 days (Supplementary Table 6), all underwent percutaneous coronary intervention (PCI) within the first week after ED admission (Supplementary Fig. 7), and among those who received stents, 36% required multiple stents (compared to 30% for nOMI).

### Model performance in internal test sets
The model performance in the random and temporal test sets is presented in Fig. 2C (discriminative performance of the super-classes of the model), Supplementary Fig. 8 (discriminative performance of the super and sub-classes of the model), and Supplementary Table 2 (metrics of both discrimination and calibration for all super- and sub-classes, including model uncertainty from non-parametric bootstrap). Supplementary Fig. 9 presents the receiver operating characteristic (ROC) and precision-recall curves across both test datasets. These above-mentioned results present metrics comparing one given class vs all the other classes (OvA) or multiclass comparisons. Supplementary Fig. 10 presents metrics for one vs one (OvO) comparisons. In the SwED test sets, the model demonstrated excellent discriminative performance for the OMI super-class, achieving an OvA C-statistic of ≥0.95. For the nOMI super-class, the model yielded an OvA C-statistic of ≥0.87. The model also performed well in identifying the OMI culprit vessel, but with variability in the results for some specific sub-classes and test splits. This applied to OvA comparisons, while the discriminative performance was poor for one OvO comparison between OMI LCX and OMI RCA. At a false positive rate of 5%, the model achieved a sensitivity of 0.87 for OMI across both SwED test sets.

In both SwED test sets combined, the model captured significantly more STEMIs compared with the automatic 12SL diagnosis statements (part of the GE MUSE data format) at the same number of false positives (Supplementary Tables 10, 11). A total of six STEMI ECGs, which were correctly called in the diagnosis statements, had a low predicted probability of STEMI from our model. A manual evaluation of these (J.S., consultant cardiologist), revealed that all except one had an unclear ST-pattern that should not result in a confident STEMI diagnosis. When it comes to missed STEMI cases, it should be noted that our dataset contains some STEMI label noise, as shown in a previous

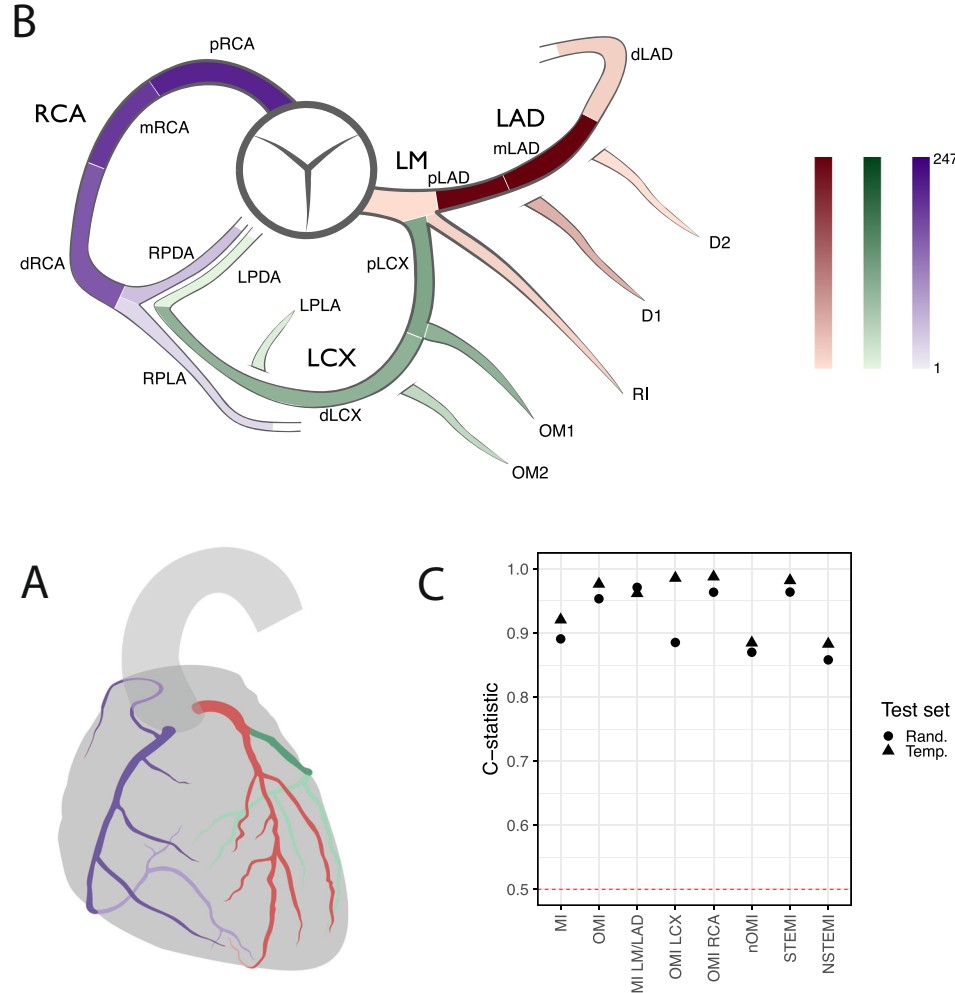

**Fig. 2 | Localization of occlusion myocardial infarctions and model performance.** Panel **A** shows an anatomical heart with the three main branches of the coronary tree, where the model tries to predict the location of an occlusion. RCA (right coronary artery) in purple, LM/LAD (left main coronary artery+left anterior descending artery) in red, and LCX (left circumflex artery) in green, where a lighter color corresponds to a coronary artery that is hidden behind the current view of the heart. Adapted from File:Coronary arteries.svg by Patrick J. Lynch (Wikimedia Commons), CC BY-SA 3.0 https://creativecommons.org/licenses/by-sa/3.0. Panel **B** shows a schematic illustration of the coronary artery segments centered around the aorta. The same color scheme is used as in Panel **A**, but the color intensity corresponds to how frequent occlusion myocardial infarctions (OMI) were in the given segment. Panel **C** shows the discriminative performance (C-statistic) when comparing a given class (x-axis) with all other classes in the random or temporal test set. The dashed horizontal line represents a random guess. Different subclasses of MI are evaluated: with/without occlusion MI (OMI/nOMI), with/without the presence of a ST-segment elevation (STEMI/NSTEMI), and OMI culprit vessel localization (RCA, LM/LAD, or LCX).

study[12]. The STEMI label was determined at discharge from the coronary care unit or emergency room when the whole care episode could be summarized. The ECGs in the test sets of this study may hence not always be the ones guiding the final diagnosis.

### External validation

We also tested model performance in the publicly available PTB-XL dataset (European), the CODE-II dataset (Brazilian; 12 regions), and the InCor dataset (Brazilian; São Paulo). The model's discriminative performance for OMI vs control was good (C-statistic of 0.83) in the InCor external validation set and with comparable results in evaluations stratified by ECG vendor (GE MUSE and Mortara) (Supplementary Table 8), and the model performance was better for OMI-STEMI than OMI-NSTEMI (0.96 vs 0.82, as shown in Supplementary Table 9). In addition, the model achieved excellent performance (Supplementary Table 3) in differentiating STEMI vs control as well as LBBB vs control (the only available labels in CODE-II and PTB-XL, relevant for this study). The control group in CODE-II mostly includes normal ECGs and the control group in PTB-XL includes normal

controls. The performance was mostly comparable across age, sex, and comorbidity strata in the CODE-II external validation set (Supplementary Tables 12, 13). Calibration plots indicated acceptable agreement between predicted probabilities and observed outcomes, as shown in Supplementary Fig. 11. The calibration plots are presented for the super-class outcomes with a sample size that gives adequate resolution in the plots.

### Robustness and subgroup analyses

To assess the robustness of our model and rule out the influence of proxy variables, we stratified performance by several potential confounders. Stratified C-statistics are presented in Supplementary Fig. 12 for the following subsets: age tertiles, sex, did the patient visit the emergency department at Karolinska Hospital (main source of the training data) or another emergency department in the Stockholm region, ECGs recorded using the most common machine type (MAC55) or not, ECGs recorded using the most common software (v237) or not, prior hospitalization due to cardiovascular disease, and prevalent left bundle-branch block (LBBB). The discriminative performance was

mostly comparable across strata, but some differences should be highlighted. The model performed better in patients without LBBB, younger patients, and patients without prior cardiovascular disease hospitalizations. Comparable results were seen across year tertiles as shown in Supplementary Fig. 13.

Some comorbidities were over- or underrepresented among certain high probability misclassifications, as shown in Supplementary Table 4. A manual review (G.M.M.P., consultant cardiologist) of CODE-II ECGs misclassified with a high probability suggested that many discrepancies were due to incorrect reference diagnoses. In several cases, the reviewer agreed more with the model than with the original label. Among the false positives, all ECGs showed ST-segment elevation, with eight displaying classic STEMI patterns and two showing features of pericarditis or a dyskinetic segment. Half of the false negatives showed clear STEMI patterns, while the rest included confounding features such as LBBB, subendocardial ischemia, left ventricular hypertrophy, premature ventricular contractions, sinus tachycardia, and atrial fibrillation with high ventricular rate, which may explain the model's low confidence (Supplementary Table 5). The false negatives with a clear ST-segment elevation had other ECG abnormalities, such as left intraventricular conduction disturbances and atrial fibrillation with a high ventricular rate.

The model displayed excellent discrimination but lower average precision. It appropriately expressed uncertainty in terms of low class probabilities in challenging cases, such as ECGs with technical noise that the normalization had not removed, in cases with very early ECGs on arrival at the ED or very late when thrombolysis treatment or even PCI may have been performed, and with possible (but not probable) STEMI in PTB-XL. A manual review (J.S., consultant cardiologist) of SwED test set patients with multiple ECGs from the same visit, where the model outputs very different probabilities was performed. The evaluation showed that the ECGs given a high OMI probability indeed had a clear infarction pattern whereas ECGs with a low probability often had issues such as a much weaker infarction pattern or major technical noise.

## Limitations

Weaknesses include no availability of external validation sets with OMI culprit localization. However, we have included three external validation sets that together capture different geographical regions, machine types, years, and labels for OMI, STEMI/NSTEMI, and LBBB. Certain subgroups remain an issue; perimyocarditis still[12] raises some confusion for the model, and discrimination of nOMI in the LBBB subgroup is poor but better than chance (Supplementary Table 4, Supplementary Fig. 12). The challenge in separating between OMIs with occlusions in LCX and RCA is not unexpected; the origin of the posterior descending artery is more often the RCA than the LCX (Fig. 1A+B). This discrimination is challenging also for humans using ECGs[14]. The number of cases in some of the outcome classes is modest (Supplementary Table 1), with quite few ECGs to learn from in training and few ECGs in evaluation leading to some uncertainty. Still, the results across outcome classes are encouraging, but some outcome classes such as the LCX culprits would likely benefit from a larger sample size in a future similar model. The CODE-II dataset lacks paired coronary angiography, so it should be noted that acute coronary occlusion cannot be confirmed for those with a STEMI label.

## Strengths and clinical implications

Strengths of this study include a strict OMI definition based solely on catheterization data that can differentiate acute from chronic coronary occlusions. OMI definitions that rely on cardiac troponin levels are suboptimal for several reasons. First, STEMI cases are rushed past the ED to the catheterization lab without stopping for cardiac troponin testing. Second, troponin levels are low in the most acute phase. Third, the acuity of less than total or near-total occlusions is unknown. Our model does not rely on troponins or any other external dependencies that may lead to delays. Most myocardial infarction deaths occur in low- and middle-income countries, where timely access to reperfusion therapy is often limited. In these settings, accurate and affordable tools such as machine learning-based ECG interpretation could play a vital role in increasing access to early diagnosis and life-saving treatment[15]. An unanticipated observation is that our model's STEMI predictions outperform a STEMI model trained on the CODE-II dataset itself, when testing it on the CODE-II dataset. This leads us to conclude that a model trained on human labels (the CODE-II data) can reach the best conceivable human level, while a model trained on objective outcome labels (our coronary catheterization data) can acquire beyond-human capabilities, even when the test set is based on human labels. Given that the number of cases in the outcome classes varies over the years, it is reassuring that our model does not only perform well in the random test set but also in the temporal test set (reflecting the last year of clinical practice in the main study sample). The model also shows promising performance for OMI, STEMI-OMI and NSTEMI-OMI in the external validation set InCor (Brazilian data collected 2017–2024 and including the separate ECG vendor Mortara).

We herein developed and validated a machine learning model for the identification and localization of OMI. The clinical usefulness of such a model can be substantial as it can (1) identify OMI patients that are in need of urgent revascularization, and avoid acute activation of the catheterization lab if there is no OMI; and (2) inform the angiographer on which is the most likely culprit, in the case of concurrent chronic occlusions[11]. This is a major development over the current STEMI/NSTEMI paradigm, as ST-elevations are only present in a fraction of OMIs in need of acute revascularization[2], leading to unnecessary delays[5,6], deaths[2], and costs. Prior machine learning models using ECG data[8,9] have shown high performance for classification of OMI. Our model reached similar or higher performance and can additionally point out the localization of the occlusion. Our findings further strengthen the notion that OMI is a potentially useful clinical entity, and the potential usefulness of machine learning models for its timely detection and localization. Because human oversight of OMI detection on the ECG is limited, randomized clinical trials with patient-relevant outcomes are needed before clinical deployment of such models.

## Methods

The study was approved by the Swedish Ethics Authority, accession numbers 2022-07108-01, 2023-05042-02, 2023-09-22, and 2024-04058-02 (SwED), and also approved by the Research Ethics Committee of the Universidade Federal de Minas Gerais, CAAE 85892325.1.0000.5149 (CODE-II) as well as the Research Ethics Committee of the Clinics Hospital, Heart Institute, University of São Paulo Medical School, CAAE 45070821.3.0000.0068 (InCor). For SwED, informed consent was waived by the Swedish Ethical Review Authority (on the basis of public interest, in accordance with GDPR Article 9(2)(j) and the Swedish Ethical Review Act [2003:460]). For CODE-II, as this was a secondary analysis of anonymized data, the Research Ethics Committee of the Universidade Federal de Minas Gerais waived the requirement for individual informed consent.

For InCor, informed consent was waived due to the retrospective nature of the study, which exclusively used previously collected and fully anonymized data extracted from electronic health records (EHR), with no direct patient contact or intervention. This approach complies with applicable ethical and regulatory frameworks. The Research Ethics Committee of the Heart Institute (InCor), Hospital das Clínicas da Faculdade de Medicina da Universidade de São Paulo, determined that the use of de-identified, retrospective data poses minimal risk to participants and does not require individual informed consent, in accordance with national regulations (e.g., CNS Resolution No. 466/2012 and complementary guidelines).

## Main study sample

The main study sample used in training and validation of the model was obtained from a subset of the Swedish Emergency Department Database (SwED), as described in a previous study of the prediction of ST-elevation and non-ST-elevation myocardial infarction (STEMI and NSTEMI)[12]. In brief, the SwED sample includes patients ≥18 years old with a Swedish personal identifier number and available ED data from multiple emergency departments in Sweden, between 2003 and 2017. The sample was linked to national registries (the patient [inpatient and specialized outpatient], prescribed drug, and death registries), national quality registries (SWEDEHEART [Swedish Web-system for Enhancement and Development of Evidence-based care in Heart disease Evaluated According to Recommended Therapies; a Swedish nation-wide quality register] sub-registries RIKS-HIA [Register of Information and Knowledge About Swedish Heart Intensive Care Admissions] and SCAAR [Swedish Coronary Angiography and Angioplasty Registry]), as well as a regional database of ECGs (Karolinska ECG database) and electronic health records.

ECG availability restricted the present study to the Karolinska ECG database, Stockholm, Sweden and the years covered by all required data sources at time of the SwED data extraction was limited to 2005-2016. The Karolinska ECG database mainly consists of patients admitted to the Karolinska Hospital, Stockholm, Sweden (85%) but also includes other Region Stockholm hospitals such as Södertälje and Norrtälje. Hence, the study sample included all adult patients who presented to the emergency department (ED) in Region Stockholm, Sweden, between 2005 and 2016, where a standard 12-lead ECG was recorded within 24 h of their visit. The inclusion and exclusion criteria used to define the study sample are described in Fig. 1, which were largely consistent with the prior study[12]. Notable differences were the requirement that all myocardial infarction (MI) cases have undergone angiography within seven days of their ED visit (in order to differentiate between OMI and nOMI as well as to determine localization), the additional inclusion of patients with prevalent left bundle branch block (LBBB; as this is a non-negligible fraction of all acute occlusions that any MI model should be able to identify), and the inclusion of controls from the years 2005–2006 (the granular MI sub-classes described below were not recorded for these years, hence only controls were included).

## Exposures and outcomes

High-quality data on exposures and outcomes were available from emergency department discharge records, electronic health records, the SWEDEHEART registry, and linked hospitalizations, with patient records joined using the Swedish personal identifier number.

The exposures included digital ECG data, age at time of the ED visit, and sex. Sex was determined using the Swedish personal identifier number. The ECG recordings were available in the GE MUSE (v9) XML format, with each lead represented by a numerical signal vector together with detailed meta-data of the recording. Standard 10-second, 12-lead ECG recordings sampled at 250–500 Hz and stored with a resolution of 4.88 μV per least significant bit in a 16-bit format (dynamic range of around ±160 mV) were used; eight leads were retained (I, II, V1–V6) as four of the standard leads are linear combinations of these and thus redundant. All ECGs had baseline wander removed, were resampled to 400 Hz, and all leads were zero-padded to a fixed length of 4096 samples.

The outcome labels are primarily defined using data from the SWEDEHEART quality registry where an overall data accuracy of around 96% has been reported from monitor visits. Further, SWEDEHEART has a coverage of 100% for Swedish patients undergoing angiography, which is an inclusion criteria for the myocardial infarction cases in the present study[16]. The study extended the three outcome classes of the prior study[12] (control, STEMI, and NSTEMI) into ten mutually exclusive MI sub-classes, incorporating additional granularity regarding OMI and culprit vessel localization (definitions described below). Additionally, controls were stratified into those with and without perimyocarditis, a subgroup previously found to be frequently misclassified as STEMI[12]. The MI sub-classes included:

1. Control with perimyocarditis
2. Control without perimyocarditis
3. NSTEMI, nOMI
4. NSTEMI, OMI, LM/LAD
5. NSTEMI, OMI, LCX
6. NSTEMI, OMI, RCA
7. STEMI, nOMI
8. STEMI, OMI, LM/LAD
9. STEMI, OMI, LCX
10. STEMI, OMI, RCA

MI was defined as a new onset acute myocardial infarction (ICD10:I21-I23 set as diagnosis) as described previously[12], and the sub-class OMI was defined as a newly formed occlusion with complete or close to complete blockage requiring more urgent intervention as opposed to old chronic occlusions/stenosis in the other MI cases that have slowly been built up (where the heart, to some extent, might have adapted, by forming collateral vessels bypassing the blockage). OMI was defined based on the SWEDEHEART SCAAR variable OCKL. The culprit vessel of the OMI was defined as the segment marked as having the newly formed occlusion, based on the characteristics of the occlusion upon guidewire maneuvering. The SCAAR database contains information on the coronary segments affected, with twenty segment labels according to a modified version of a previous definition[17]. The fraction of MI cases with an occlusion in some of these segments is very small, and it is sufficient to inform the angiographer on which is the most likely culprit; therefore the segments were grouped into the three main branches: left main coronary artery together with the left anterior descending artery and the intermediary (LM/LAD), left circumflex artery (LCX), and right coronary artery (RCA) as illustrated in Fig. 2, which shows all segments together with the main branches.

Instead of excluding patients with prevalent LBBB as in the STEMI/NSTEMI study, the presence of LBBB was incorporated as a separate outcome label that could co-occur with the ten MI classes. LBBB classification was based on the LBBB predictions from an external, machine learning model trained on Brazilian data, which has been shown to outperform cardiology resident medical doctors in the classification of LBBB from ECGs[18]. The model predictions were supplemented with prevalent ICD10:I44.6-I44.7 diagnoses from the patient registry and SWEDEHEART annotations of LBBB, i.e. any diagnosis set up until the ED visit from all available diagnosis data of the patient. A patient was classified as having an LBBB at the ED visit if all ECGs of the ED visit had a Pr(LBBB) ≥ 0.5 (cutoff selected by visual inspection of Supplementary Fig. 2) or if a prevalent LBBB diagnosis was present in any of the other data sources. The exact type of LBBB (incomplete or complete) is not always known in the Swedish patient registry data due to inexact ICD10-SE codes set. However, for patients with an exact ICD10-SE code set, 97% were complete LBBB (ICD10-SE:I44.6A). In the Brazilian model used to complement our labels, the outcome was complete LBBB only, hence our LBBB definition corresponds to complete LBBB.

## Outcome combinations

For the ten non-overlapping MI outcome sub-classes, several super categories were of particular interest:

- OMI = combination of all OMI classes
- OMI LM/LAD = combination of all OMI classes with LM/LAD as the culprit vessel
- OMI LCX = combination of all OMI classes with LCX as the culprit vessel

- OMI RCA = combination of all OMI classes with RCA as the culprit vessel
- MI = combination of all MI classes
- NSTEMI = combination of all NSTEMI classes
- STEMI = combination of all STEMI classes

The ability to predict each individual OMI case – along with their location – is paramount in order to inform the angiographer on which is the most likely culprit in high urgency patients. Compared to the previous STEMI/NSTEMI study, predicting all MI cases not only has clinical applications but also helps evaluating model performance in test sets where only MI (yes/no) labels are available.

### Data splitting

Data were split into 70% training, 10% validation, and 20% test sets, where a patient could only be included in one of these sets. The test set was further divided into a random test set and a temporal test set, with the latter including patients whose first visit occurred in the final year (≥2016-01-01) of the study period, which is further described in Fig. 1. To augment the training data, multiple ECGs recorded at the same ED visit of a patient were used. In contrast, only one ECG per ED visit of a patient was used for validation and testing, and if multiple ECGs had been recorded at the ED visit for patients in the validation or test set, only one of these ECGs were retained at random.

### Model architecture and training

The model was based on a convolutional neural network (CNN) using a residual network (ResNet) architecture with skip connections, adapted for one-dimensional ECG signals. The model architecture has been described previously[18], and the present study extends that previous model architecture. In addition to the ECGs, age and sex were used as predictors (age was mean-centered and re-scaled to unit variance using mean = 61.9 and sd = 19.5). Age and sex were first mapped to a higher-dimensional space ($n = 64$), through a fully connected linear layer using ReLU activation to capture non-linear relationships. Age, sex, and ECGs were then concatenated before the final fully connected prediction layer. As in the STEMI/NSTEMI study, an ensemble of five independently trained model members was used, with their logits averaged to generate final predictions, i.e. output predicted probabilities of the MI sub-classes, control sub-classes, and LBBB, as diagnosis at time of the ECG. The same hyperparameters as in the STEMI/NSTEMI study were used. Besides the weighted loss described below, the model architecture of the present study is identical to that of the STEMI/NSTEMI study[12].

The model was trained using cross-entropy loss for the MI sub-classes and binary cross-entropy loss was used for LBBB. These losses were combined as: $weight_{CE}*loss_{CE} + weight_{BCE}*loss_{BCE}$, where $weight_{CE} + weight_{BCE} = 1$. A hyperparameter search was conducted for determining the optimal BCE:CE weight ratio while all other hyperparameters were held fixed. The BCE:CE weight ratio of 0.3 yielded the best performance trade-off between MI sub-class and LBBB classification (Supplementary Fig. 3). The model loss was minimized over up to 150 epochs (Supplementary Fig. 4).

### Model validation sets

The model was primarily evaluated in the SwED random and temporal test set described above, drawn from the same database used to train the model, including all outcomes which the model was trained to predict. Further, two external validation sets (PTB-XL and CODE-II) including labels only for STEMI vs not STEMI were used to evaluate the model's discriminative performance for STEMIs. Finally, the InCor test dataset was included to evaluate the model's performance for OMI and STEMI/NSTEMI.

The PTB-XL is a publicly available European database of 21,837 10-second 12-lead ECGs annotated with 71 different ECG statements,

including cases of myocardial infarction[19,20]. From PTB-XL, 63 acute myocardial infarction cases with a confirmed ST-elevation, 279 LBBBs and 197 randomly selected normal controls without ST-elevation were manually curated from the PTB-XL database. The inclusion criteria of the PTB-XL external validation set used in this study is summarized in the Supplement.

The CODE-II dataset[21] is an updated version of the first CODE dataset[18], with exams collected and annotated using significant improvements in the internal operational system of the Telehealth Network of Minas Gerais (TNMG), Brazil. This system ensures standardized, high-quality ECG diagnostic classes. The dataset comprises 2,735,269 exams from 2,093,807 unique patients, recorded between January 2019 and December 2022. For this study, only the first exam from each patient was included, resulting in a total of 2,093,807 exams. The dataset primarily includes patients from primary care centers, along with some from hospitals, emergency departments, and ambulances. The median age is 54.2 years (IQR: 40.9, 66.3), with 40.9% of the patients being male. Among the 66 ECG CODE diagnostic classes, there is one comparable to STEMI (2735 exams, 0.1%) and LBBB (40,380 exams, 1.9%) diagnostic classes, with the remaining classes aggregated as not STEMI (2,081,141 exams, 99.4%) for this study. Overlap is observed between exams labeled as LBBB with STEMI (56 exams) and not STEMI (30,393 exams).

The InCor-OMI external validation set comes from a single-center cohort with 10 s 12-lead ECGs acquired from patients presenting to the emergency department of the Heart Institute (InCor) of the University of São Paulo Medical School, Brazil. A total of 401 ECG exams recorded between 2017 and 2024 were included for evaluation of this study. The median age was 67.0 years (IQR: 57.0, 76.0), and 63.3% of the patients were male. Each exam is labeled for the presence of occlusion myocardial infarction by manual review of emergency clinical notes and coronary angiography reports by a trained cardiologist. Patients were included in the OMI group if they underwent coronary catheterization during the ED admission and had at least one culprit coronary lesion with ≥95% obstruction. Controls free from OMI were randomly sampled from ED patients admitted during the same period as the OMIs who did not undergo coronary catheterization at any point during their hospital stay. In total, 41.1% OMI were included (Supplementary Table 7) and these were further annotated as STEMI or NSTEMI by a cardiologist who assessed the presence of a ST-elevation. ECGs were acquired from two vendors: Mortara (70.8%) and GE MUSE (29.2%).

To better understand the model's classification errors in the CODE-II dataset, we manually reviewed 20 ECGs that were misclassified by the model. This set included 10 false positives, defined as cases not labeled as STEMI where the model predicted a STEMI probability greater than 0.9, and 10 false negatives, where STEMI cases received a predicted probability less than 0.1. The ECGs were reviewed by a consultant cardiologist (G.M.M.P.), who was aware of the predicted probability.

We also compared the model's prediction of STEMI versus the GE MUSE 12SL automatic diagnosis statements as described in the Supplement.

### Metrics of discrimination and calibration

The C-statistic and average precision were calculated to evaluate the model's discriminative performance. These metrics were primarily assessed using one vs all other (OvA) comparisons, where each class was evaluated against all other classes combined, resulting in the 'rest' group being dominated by controls without myocarditis or pericarditis. Additionally, C-statistics were also computed using one vs one (OvO) comparisons, where each class was directly compared to another specific class.

As metrics of model calibration, the Brier score and the Expected Calibration Error (ECE) were calculated. The Brier score was calculated as the mean squared error on the probability scale, both in binary OvA

calculation but also as multiclass calculation, averaging the OvA Brier scores across all classes. The Expected Calibration Error (ECE) was computed as the weighted average of the absolute differences between the accuracy and the predicted confidence within 10 probability bins. Confidence for each prediction was defined as the maximum predicted probability across classes, reflecting the model's certainty in its top choice.

We visualized the calibration of our model using continuous calibration plots of the observed case frequency vs the average predicted probability. We used the combined random and temporal test set for these plots to increase the total number of cases and calibration resolution. Model calibration was assessed using both a logistic calibration intercept and slope and a flexible smooth calibration curve estimated via a penalized thin-plate regression spline within a generalized additive model.

Confidence intervals (95%) were calculated by non-parametric bootstrapping using the percentile method on 2000 draws, unless otherwise stated. The bootstrap sampling was stratified so that each draw included the same number of cases as in the original data.

### Model performance in subgroups

We evaluated whether the model made incorrect predictions (with a high probability) for any of the recorded comorbidities of the patients. For each outcome class, we selected all records with a predicted probability of the class >0.5. Among these, we compared correctly versus incorrectly classified patients based on the true class labels. A two-sided non-parametric conditional independence test based on the maximum test statistic (via the coin package) was then used to assess whether specific 3-character ICD-10 diagnosis codes were over- or underrepresented among the misclassified patients vs correctly classified patients. All available 3-character ICD-10 codes were tested, and those with a false discovery rate (FDR) < 0.01 were highlighted.

In a manually selected list of subgroups, where the model performance might differ, we calculated the C-statistic in OvA comparisons. Given that some subgroups are small, we used the combined random and temporal test set in SwED for the groups control, nOMI, and OMI.

### Reporting summary

Further information on research design is available in the Nature Portfolio Reporting Summary linked to this article.

## Data availability

The SwED data are available from the Swedish Board of Health and Welfare and the included healthcare regions, but restrictions apply to the availability of these data, which were used under license for the current study and cannot be publicly shared. This license does not allow data sharing, to protect patient privacy. Contact the corresponding author (johan.sundstrom@uu.se) for any questions related to how to request the data from the original data holders (the healthcare regions and health authorities). Such requests require ethical approval from the Swedish Ethics Authority and are expected to take several months to process. The PTB-XL database is a publicly available resource[19,20], which was downloaded from https://doi.org/10.13026/zx4k-te85 for the present study. The full CODE-II data cannot be made publicly available, in accordance with the principle of minimizing risks that come with unrestricted direct access. Smaller datasets with limited details on patients will be made available upon publication[21]. The data is available for non-commercial purposes, for research or educational institutions. Data access requests should be forwarded to the telehealth center of Minas Gerais (telessaude.hc-ufmg@ebserh.gov.br), with antonio.horta.ribeiro@it.uu.se in copy. The request will be processed within one month. We recommend double-checking the original CODE-II publication for additional information. The anonymized InCor dataset consists of personal data from Brazilian individuals and is subject to the Lei Geral de Proteção de Dados (LGPD), which establishes strict requirements for data protection, privacy, and governance. It cannot be shared publicly for privacy reasons. Requests for access to InCor-derived data for non-commercial purposes by research or educational institutions will be evaluated on a case-by-case basis. Requests must be submitted to Prof. Jose Eduardo Krieger (j.krieger@hc.fm.usp.br) and include: (i) the study protocol, (ii) institutional affiliation of the investigators, (iii) ethics approval or waiver, when applicable, (iv) a data protection plan, and (v) a detailed description of the requested variables and intended analyses. All requests will undergo institutional and ethical review, with an expected response time of up to two months, subject to applicable review processes. If approved, access will be granted under a formal Data Use Agreement (DUA), which prohibits data sharing, commercial use, attempts at re-identification, and unauthorized linkage with external datasets, in compliance with applicable data protection regulations.

## Code availability

All code is available at https://github.com/stefan-gustafsson-work/omi and parameter/hyperparameter estimates of the presented models are available upon request from the corresponding author.

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

## Acknowledgements

The inclusion criteria were kept as wide as possible: only minors and patients without a personal identification number were excluded. The patients or public had no role in the design of the study. We thank all the patients whose data contributed to the development and evaluation of this prediction model. The computations were enabled by resources in project sens2020005 and sens2020598 provided by the Swedish National Infrastructure for Computing (SNIC) at UPPMAX. A.H.R. is partially supported by the eSSENCE strategic collaborative research programme for the research of this work. D.G. was funded by the University of Tübingen and Boehringer Ingelheim AI & Data Science Fellowship Program for the research of this work. P.E.O.G.B.A. is a CNPq scholarship holder. A.L.P.R. is supported in part by CNPq (National Council for Scientific and Technological Development, grants 310790/2021-2, 409604/2022-4, 445011/2023-8, and 408659/2024-6), FAPEMIG (Minas Gerais State Foundation for Research Support, grant RED 00192-23) is a member of the CIIA-S (Innovation Center on Artificial Intelligence for Health), and the IATS-CARE (Institute for Health Assessment and Translation for Chronic and Neglected Diseases of High RElevance). The study was funded by The Kjell and Märta Beijer Foundation, Anders Wiklöf, the Wallenberg AI, Autonomous Systems and Software Program (WASP) funded by Knut and Alice Wallenberg Foundation, and Uppsala University. This project has received funding from the European Research Council (ERC) under the European Union's Horizon Europe research and innovation programme (grant agreement n° 101054643). The InCor-OMI external validation set and the computational infrastructure were supported by FAPESP (São Paulo State Foundation for Research Support, grants #2025/27076-4 and #2024/13328-9). Open access funding was provided by Uppsala University. The funders had no role in study design; collection, analysis, and interpretation of data; writing of the report; or decision to submit the paper for publication.

## Author contributions

S.G: Conceptualization, Data Curation, Methodology, Formal Analysis, Visualization, Writing – Original Draft, A.H.R: Methodology, Writing – Review & Editing, D.G: Methodology, Writing – Review & Editing, P.E.O.G.B.A: Validation, Visualization, Writing – Review & Editing, N.P.: Methodology, Writing – Review & Editing, G.M.M.P: Validation, Investigation, Writing – Review & Editing, M.A.G: Validation, Writing – Review & Editing, J.E.K: Validation, Writing – Review & Editing, F.M.D: Validation, Writing – Review & Editing, A.L.P.R: Validation, Writing – Review & Editing, D.L: Visualization, Writing – Review & Editing, T.B.S: Supervision, Funding Acquisition, Methodology, Writing – Review & Editing, J.S: Project Administration, Supervision, Funding Acquisition, Conceptualization, Methodology, Writing – Original Draft.

## Funding

## Competing interests

The authors declare the following competing interests: J.S. reports direct or indirect stock ownership in companies (Anagram kommunikation AB, Sence Research AB, Symptoms Europe AB, MinForskning AB) providing services to companies and authorities in the health sector including Amgen, AstraZeneca, Bayer, Boehringer, Eli Lilly, Gilead, GSK, Göteborg University, Itrim, Ipsen, Janssen, Karolinska Institutet, LIF, Linköping University, Novo Nordisk, Parexel, Pfizer, Region Stockholm, Region Uppsala, Sanofi, STRAMA, Takeda, TLV, Uppsala University, Vifor Pharma, WeMind. AHR reports acting as consultant to Einthoven Technologia LTDA, with vesting options. The remaining authors declare no competing interests.
