## [Transparent Peer Review file · Nature Communications]

A deep learning ECG model for identification and localization of occlusion myocardial infarction

Corresponding Author: Professor Johan Sundstrom

Version 0:

Reviewer comments:

Reviewer #1

(Remarks to the Author)

The manuscript and its supplement are well written and comprehensively describe the development and testing of a deep neural network (DNN) designed to detect OMI/nOMI/STEMI/NSTEMI and discriminate the culprit coronary vessel. By including known confounders such as LBBB and perimyocarditis, the authors strengthen model performance and clinical robustness. The study tackles a highly relevant problem—localization of occlusion myocardial infarction using ECG-based deep learning—which is clinically valuable given that OMI is often delayed or missed and accurate localization can guide urgent management. Its strengths include the use of a large, real-world dataset (SwED), multi-class classification extending beyond traditional STEMI/NSTEMI differentiation, and external validation across two international datasets (CODE-II and PTB-XL). However, while impactful, the novelty is somewhat limited by prior work on AI-based myocardial infarction detection from ECGs; the manuscript's primary contribution lies in its fine-grained vessel-level localization (LM/LAD, LCX, RCA) and the inclusion of diagnostically challenging subtypes (nOMI, perimyocarditis).

Review questions:

- What are the noteworthy results?

The likely culprit coronary artery is classified for not only STEMI ECGs but OMI and NSTEMI ECGs as well

- Will the work be of significance to the field and related fields?

Yes, the work is significant

- Does the work support the conclusions and claims, or is additional evidence needed?

The work supports the conclusions.

- Are there any flaws in the data analysis, interpretation and conclusions? - Do these prohibit publication or require revision?

There are no flaws in data analysis.

- Is the methodology sound? Does the work meet the expected standards in your field?

The methodology is sound.

- Is there enough detail provided in the methods for the work to be reproduced?

There is enough detail to reproduce a similar deep learning model.

General comments:

1. The authors should be praised for including the effect of LBBB and perimyocarditis in the model, but why not include other confounders such as left ventricular hypertrophy and early repolarization? Why not make perimyocarditis an independent output like LBBB?

2. There was a comment on the trend of more NSTEMI cases treated in the cath lab over the study period. Does this represent a change in clinical practice that could mean a selection bias exists in earlier cases compared to later cases? I suggest adding something that reminds the reader that even if that bias exists, the final year test set would mean testing occurs on the most recent clinical practice.

Specific comments:

1. Supplement, table 1: the number of cases in many of the OMI subgroups are small, be sure to mention this as a study limitation

2. Supplement, table 3: why does the table have an NA entry for LBBB in PTB-XL? LBBB annotations should be available in PTB-XL
3. Supplement, figure 1, panel B: change 80% in training to 70% to match what is found in the text
4. Supplement, figure 6: do you have a hypothesis for large difference in test performance for OMI, NSTEMI, LCX group? Consider mentioning the hypothesis as something to follow up on for future research. Could it be driven by a change in clinical practice over time to send more LCX STEMI equivalent to cath lab?
5. Supplement, figure 9: is binned calibration plot considered common knowledge? If not, please include a reference
6. "Of these ECGs, 1,583 (0.3%), 4,279 (0.8%), and 534,510 (98.9%) cases were labeled at time of coronary catheterization by the coronary angiographer as OMI, nOMI (myocardial infarction not classified as OMI), and control (no myocardial infarction), respectively." This sentence could be misinterpreted to mean that 500,000 controls were labeled as part of coronary catheterization.
7. "...newly formed coronary total occlusion (TIMI blood flow 0)." Is there any guidance, explanation or reference that can be supplied to explain how "newly formed" is determined?
8. Severe imbalance (OMI only 0.3% of ECGs) likely affects rare subclass performance, e.g., NSTEMI-OMI-LCX has very low AP despite good C-statistics. What specific measures were taken to account for this?
9. As mentioned above, it is not clear how the random and temporal test sets were created.
10. The term "prevalent LBBB" is used several times. What does prevalent mean in this context?
11. Supplement, table 4: What is the reader expected to do with table 4? Are we expected to know the codes or are we expected to look up the codes or are we just expected to see repeated codes? I don't see how table 4 helps the reader understand the algorithm mistakes without looking up every listed code.
12. "Half of the false negatives showed clear STEMI patterns..." This would be troubling to a clinical user. Is there any explanation that could help give the clinical user confidence that clear STEMI will not be missed?
13. "...perimyocarditis still raises some confusion for the model..." why not make perimyocarditis another model output like LBBB?
14. Conclusion: "...guide the angiographer to the critical coronary branch, in the case of concurrent chronic occlusions" Is the precision high enough for the cath lab user to confidently stent the predicted culprit first without a complete survey of coronary artery tree? This conclusion seems a bit strong.
15. "OMI was defined based on the SWEDEHEART SCAAR variable OCKL. The culprit vessel of the OMI was defined as the segment marked as having the newly formed occlusion." Is the newly formed occlusion a very confident diagnosis? If there is doubt, it should be mentioned in the study limitations.
16. Outcome combinations: Is it correct that the MI label means acute rather than old?
17. Data splitting: Just a comment – it would be interesting to see the performance if a human chose the best ECG for a patient rather than a random ECG. This would mimic the paramedics when they have the right constellation of symptoms.
18. Model validation sets: It is not clear what "manually curated" means for PTB-XL ECG choice. Does it mean that there were more than 75 acute MI cases but 75 were chosen?
19. Model performance in subgroups: it would be helpful to indicate which tables and figures represent the text. The reader can go back and forth to determine which figure/table but I suggest adding references to make it easier for the reader.
20. No detailed breakdown of performance by sex, ethnicity, or comorbidity burden in external data.
21. No use of saliency maps, Grad-CAM, or lead-level attribution to understand model focus. Clinicians will require interpretability to build trust and integrate into workflows—especially for localization claims.
22. Manual re-review (Table 5) reveals that many "false positives" were actually STEMI cases upon expert review, suggesting label noise in the reference standard—this may underestimate true performance.
23. Performance across device types (MAC55 vs others) and software versions varied modestly, suggesting potential site- or vendor-specific tuning effects. This should be noted.

(Remarks on code availability)

Reviewer #2

(Remarks to the Author)

(Remarks on code availability)

Reviewer #3

(Remarks to the Author)

This studies extends the DNN models developed for prediction/diagnosis of myocardial infarction in emergency department patients (PMID: 36380048). In a large number of ECGs (N=540,372) from the Swedish Emergency Department Database paired with catheterization outcomes a deep learning model was trained and validated for the diagnosis of (non-) occlusion myocardial infarctions reaching a C-statistic of >0.95 for occlusion myocardial infarction, >0.87 for non-occlusion myocardial infarction.

High constant performance across subgroups was observed, in particular different ECG hardware/software. Even on left

bundle branch block ECGs the model performed well.

Specific comments:

- More emphasis could be placed on nOMI, a more common condition and often more difficult to decide on a culprit lesion.
- How much better does the current model perform compared to existing, automated STEMI diagnosis by the ECG machine algorithms?
- The authors select important potential confounders. Should medication intake also be considered as a confounding variable (e.g. digitalis)?
- The worse performance for LCX/RCA localization is not specific to the model, but also seen for human diagnosis. It is based on anatomical reasons because the ECG mirrors the occlusion location which can be a large RCA or an LCX depending on the vessel size and epicardial course of the artery. The authors may want to comment on this and the ability of future and different models to overcome this limitation.
- Were posterior leads (v7-9) available that could have led the model to identify LCX OMI more easily?
- The findings should be set into the context of recent studies such as Cho KH Heart J Digit Health. 2025, PMID: 40703113, Büscher A Eur Heart J. 2025 PMID: 40156923.
- Is there a potential for bias if patients who did not undergo catheterization could have missed as nOMI or, less likely, OMI?
- Whereas the baseline characteristics of the cohort are important they could be moved to the supplement if this would allow for more data on the central results in the main manuscript. Currently, the only important results presentation in the context of this study is Figure 1, panel C.
- The legend of the figure could explain nOMI vs. NSTEMI, OMI vs. STEMI.
- As the authors state, the value of the model is in the early/preclinical phase to identify patients with high risk of OMI. In this setting the exact localization of the occluded artery probably is not of high importance. They might explain in more detail what the benefit of knowing the occlusion site in OMI in the cathlab is? Both vessels are examined routinely.
- In clinic, lead III plays an important role for the diagnosis of STEMI, would this lead not add additional information?
- Since it is central for the understanding, please explain the definition of OMI (SWEDEHEART SCAAR variable OCKL) and STEMI in more detail. Does "old chronic occlusions" refer to "chronic total occlusions"? How many old chronic occlusions were observed? Is a differentiation for nOMI into type 1 and type 2 myocardial infarction possible?
- In the baseline table it might be of interest to know how many stents were placed.

(Remarks on code availability)

Version 1:

Reviewer comments:

Reviewer #1

(Remarks to the Author)

All comments and suggestions were taken into account by the authors.

(Remarks on code availability)

Reviewer #2

(Remarks to the Author)

(Remarks on code availability)

The authors have put good effort into their revisions.

I do not have further comments at this time and defer to my primary reviewer.

Reviewer #3

(Remarks to the Author)

Within the limitations of the study the authors have carefully addressed the comments made by this reviewer.

(Remarks on code availability)

Response to reviewers

We thank all reviewers for their valuable comments which we believe have significantly improved the paper. We have tried to update the manuscript where possible, unless we don't have suitable data to answer a given comment.

In addition to the updates related to the reviewers' comments, we have made minor updates of the manuscript to make it fully compatible with the TRIPOD-AI guidelines (e.g. updated title). We have also made a major update by including an additional external validation set (InCor) to externally validate model performance for the primary outcome OMI. We believe this addition greatly strengthens the manuscript. Because of the addition of a new external validation set, three new co-authors have been added. All changes are highlighted with a red font colour throughout.

We first summarize the addition of the added external validation set.

Added InCor test dataset

Update of methods:

The InCor-OMI external validation set comes from a single-center cohort with 10-second 12-lead ECGs acquired from patients presenting to the emergency department of the Heart Institute of the University of São Paulo (InCor), Brazil. A total of 401 ECG exams recorded between 2017 and 2024 were included for evaluation of this study. The median age was 67.0 years (IQR: 57.0, 76.0), and 63.3% of the patients were male. Each exam is labeled for the presence of occlusion myocardial infarction by manual review of emergency clinical notes and coronary angiography reports by a trained cardiologist. Patients were included in the OMI group if they underwent coronary catheterization during the ED admission and had at least one culprit coronary lesion with $\geq 95\%$ obstruction. Controls free from OMI were randomly sampled from ED patients admitted during the same period as the OMIs who did not undergo coronary catheterization at any point during their hospital stay. In total, 41.1% OMI were included (Supplementary table 7) and these were further annotated as STEMI or NSTEMI by a cardiologist who assessed the presence of an ST-elevation. ECGs were acquired from two vendors: Mortara (70.8%) and GE MUSE (29.2%).

Supplementary Table 7. Number of ECGs per outcome class in the InCor external validation set (São Paulo, Brazil), stratified by ECG vendor and overall. All OMI patients were further subclassified as either OMI-STEMI or OMI-NSTEMI.

Outcome	GE MUSE	Mortara	Total
Control	92	144	236
OMI	25	140	165
OMI-STEMI	1	28	29
OMI-NSTEMI	24	112	136

Update of results:

We also tested model performance in the publicly available PTB-XL test dataset (European), the CODE-II test dataset (Brazilian; 12 regions), and the InCor test dataset (Brazilian; Sao Paulo). The model's discriminatory performance for OMI vs not MI was good ($C=0.83$) in the InCor external validation set and with comparable results in evaluations stratified by ECG vendor (GE MUSE and Mortara) (Supplementary Table 8), and the model performance was better for OMI-STEMI than OMI-NSTEMI (Supplementary Table 9). In addition, the model achieved excellent performance (Supplementary Table 3) in differentiating STEMI vs control as well as LBBB vs control (the only available labels in

CODE-II and PTB-XL relevant for this study). The control group in CODE-II mostly includes normal ECGs and the control group in PTB-XL includes normal controls.

Supplementary Table 8. Metrics of discriminatory performance for OMI in the InCor external validation test (São Paulo, Brazil). OMI is compared vs not MI, stratified by ECG type (GE MUSE and Mortara) as well as overall. In InCor, 82% of the OMIs were NSTEMI, compared with 27% NSTEMI in SwED test set. Arrows indicate the direction of better performance. The C-statistic and average precision (AP) are provided.

Metric	Outcome	GE MUSE	Mortara	Total
C-statistic (↑)	OMI	0.8504	0.8277	0.8291
AP (↑)	OMI	0.6132	0.8431	0.7942

Supplementary Table 9. Metrics of discriminatory performance for OMI-STEMI and OMI-NSTEMI in the InCor external validation test (São Paulo, Brazil). OMI-STEMI and OMI-NSTEMI are compared vs not MI, stratified by ECG type (GE MUSE and Mortara) as well as overall. Only one OMI-STEMI was available for GE MUSE, so performance was not assessed for that outcome and vendor. Arrows indicate the direction of better performance. The C-statistic and average precision (AP) are provided.

Metric	Outcome	GE MUSE	Mortara	Total
C-statistic (↑)	OMI-STEMI	NA	0.9593	0.9591
	OMI-NSTEMI	0.8288	0.8265	0.8201
AP (↑)	OMI-STEMI	NA	0.8874	0.8573
	OMI-NSTEMI	0.5244	0.8020	0.7123

Response to reviewers' comments

Reviewer #1 (Remarks to the Author):

The manuscript and its supplement are well written and comprehensively describe the development and testing of a deep neural network (DNN) designed to detect OMI/nOMI/STEMI/NSTEMI and discriminate the culprit coronary vessel. By including known confounders such as LBBB and perimyocarditis, the authors strengthen model performance and clinical robustness. The study tackles a highly relevant problem—localization of occlusion myocardial infarction using ECG-based deep learning—which is clinically valuable given that OMI is often delayed or missed and accurate localization can guide urgent management. Its strengths include the use of a large, real-world dataset (SwED), multi-class classification extending beyond traditional STEMI/NSTEMI differentiation, and external validation across two international datasets (CODE-II and PTB-XL). However, while impactful, the novelty is somewhat limited by prior work on AI-based myocardial infarction detection from ECGs; the manuscript's primary contribution lies in its fine-grained vessel-level localization (LM/LAD, LCX, RCA) and the inclusion of diagnostically challenging subtypes (nOMI, perimyocarditis).

Review questions:

- What are the noteworthy results?

The likely culprit coronary artery is classified for not only STEMI ECGs but OMI and NSTEMI ECGs as well

- Will the work be of significance to the field and related fields?

Yes, the work is significant

- Does the work support the conclusions and claims, or is additional evidence needed?

The work supports the conclusions.

- Are there any flaws in the data analysis, interpretation and conclusions? - Do these prohibit publication or require revision?

There are no flaws in data analysis.

- Is the methodology sound? Does the work meet the expected standards in your field?

The methodology is sound.

- Is there enough detail provided in the methods for the work to be reproduced?

There is enough detail to reproduce a similar deep learning model.

General comments:

1. The authors should be praised for including the effect of LBBB and perimyocarditis in the model, but why not include other confounders such as left ventricular hypertrophy and early repolarization? Why not make perimyocarditis an independent output like LBBB?

Adding additional comorbidities is an excellent suggestion and something we have considered. We believe our labels for LBBB and perimyocarditis, based on ICD10-SE diagnosis codes in combination with the Brazilian model predictions, are adequate. Unfortunately, it's harder to define LVH and early repolarization in our Swedish data. Especially early repolarization we don't have any good labels for. Adding additional outcomes is actually something we are working on in a future project, but that requires good labels, which we don't have yet but are working on obtaining.

As the model is currently set up, perimyocarditis does not co-occur with MI. Instead it is set as one of the control groups. In the outcome definition we write:

Additionally, controls were stratified into those with and without perimyocarditis, a subgroup previously found to be frequently misclassified as STEMI.

The main reason for this is that perimyocarditis almost never co-occurs with MI in our Swedish data (11 ECGs with co-occurrence in the entire study database, which is probably mostly label noise) whereas LBBB co-occurs quite frequently with MI. In the rare case of perimyocarditis co-occurrence we treat such cases as MI only. Hence, the model sees perimyocarditis as a non-MI control group that is challenging as it often has a similar ECG pattern as STEMI. Perimyocarditis in patients free from MI, leading to misclassification, is the main challenge of perimyocarditis we are trying to address.

2. There was a comment on the trend of more NSTEMI cases treated in the cath lab over the study period. Does this represent a change in clinical practice that could mean a selection bias exists in earlier cases compared to later cases? I suggest adding something that reminds the reader that even if that bias exists, the final year test set would mean testing occurs on the most recent clinical practice.

Yes, we write:

*The number of samples in each sub class increased over the years, likely due to multiple factors, e.g. including expanded ECG data coverage and a **growing proportion of NSTEMI patients managed invasively (with available angiography).***

The time trends seen in e.g. Suppl. Fig. 4 are explained by multiple factors such as increasing ECG data coverage in the ECG database used, but also differences in clinical practice. There has been a substantial change in the age of the patients undergoing angiography over time due to the efficacy of coronary procedures expanding the use of them in elderly patients. Further, diagnostic procedures have changed with the introduction of high-sensitive troponin assays, resulting in more NSTEMIs over time. The increasing number of NSTEMI and elderly patients receiving angiography is described in e.g. the SWEDEHEART Annual report 2024, figure 55 (<https://www.ucr.uu.se/swedeheart/dokument-sh/arsrapporter-sh/arsrapporter-sh-aldre/01-swedeheart-annual-report-2024-english-2/viewdocument/3703>). We extended the original sentence and added this reference.

The number of samples in each sub class increased over the years, likely due to multiple factors, e.g. including expanded ECG data coverage and **changes in clinical care such as a growing proportion of non ST-elevation myocardial infarction (NSTEMI) and elderly patients with available angiography, as shown in the SWEDEHEART Annual rapport 2024.[REF]**

The reviewer raises an important issue in that such temporal trends may confound the predictions, e.g. driven by a difference in outcome proportions of the years and potentially a difference in the ECG recordings over time (e.g. different machines, software used over time).

To evaluate if the temporal trends in the number of patients in the different outcome classes affects the model predictions, we included several sensitivity analyses such as the random and temporal test set (e.g. Supplementary Table 2, Supplementary Figure 6) and analyses stratified by technical covariates (Supplementary Figure 10). When taking the uncertainty of the evaluation metric in the test set into account by non-parametric bootstrap, the model seems to perform quite similarly over time and by technical subgroup. We also added Supplementary Figure 14 (below) for additional stratification by year. In addition, the model performs well in external validation sets (PTB-XL, CODE-II, and the newly added InCor). In conclusion, the bias should be minimal, if any. Since the model performs very well (best) in the temporal test set (reflecting most recent clinical practice in available Swedish data as the reviewer points out), we expect it to do well in the classification of future cases. The InCor test, including data from another country (Brazil), different years covered (2017-2024), and different machine type (Mortara) shows promising results which further strengthens this argument.

We added this additional stratified analysis:

Supplementary Figure 14. Discriminatory performance (C-statistic) of in both SwED test sets combined, stratified by year bins. 95% confidence intervals calculated from 2000 bootstrap draws.

In the discussion we also add:

Given that the number of cases in the outcome classes varies over the years, it is reassuring that our model does not only perform well in the random test set but also in the temporal test set (reflecting the last year of clinical practice in the main study sample). The model also shows promising performance for OMI, STEMI-OMI and NSTEMI-OMI in the external validation set InCor (Brazilian data collected 2017-2024 and including separate ECG vendor Mortara).

Specific comments:

1. Supplement, table 1: the number of cases in many of the OMI subgroups are small, be sure to mention this as a study limitation

We agree that this should be highlighted as a limitation both in the training of the model as well as in the evaluation. We added:

Certain subgroups remain an issue; perimyocarditis still¹² raises some confusion for the model, and discrimination of nOMI in the LBBB subgroup is poor but better than chance. **The challenge in separating between OMIs with occlusions in LCX and RCA is not unexpected; the origin of the posterior descending artery is more often the RCA than the LCX (Fig 1. Panel A+B). This discrimination is challenging also for humans using ECGs.[PMID: 19706635]** The number of cases in some of the outcome classes is modest (Supplementary Table 1), with quite few ECGs to learn from in training and few ECGs in evaluation leading to some uncertainty. Still, the results across outcome classes are encouraging, but some outcome classes such as the LCX culprits would likely benefit from a larger sample size in a future similar model.

2. Supplement, table 3: why does the table have an NA entry for LBBB in PTB-XL? LBBB annotations should be available in PTB-XL

We thank the reviewer for this suggestion. LBBB is indeed available in PTB-XL so we have now added complete LBBB in our evaluation. Supplementary Table 3 has now been updated. We have also clarified the LBBB definition:

A patient was classified as having an LBBB at the ED visit if all ECGs of the ED visit had a $\text{Pr}(\text{LBBB}) > 0.5$ (cutoff selected by visual inspection of Supplementary Figure 11) or if a prevalent LBBB diagnosis was present in any of the other data sources. **The exact type of LBBB (incomplete or complete) is not always known in the Swedish patient registry data due to inexact ICD10-SE codes set. However, for patients with an exact ICD10-SE code set, 97% were complete LBBB (ICD10-SE:I44.6A). In the Brazilian model used to complement our labels, the outcome was complete LBBB only, hence our LBBB definition corresponds to complete LBBB.**

As for the PTB-XL ECGs included in the previous version of this manuscript, we included ECGs that had been validated by at least one cardiologist and with no electrode problems reported. All such ECGs labeled with CLBBB (complete LBBB) were added (n = 279) to the 63 STEMIs and 197 controls. There were no available records labeled with both LBBB and a Stadium I STEMI in PTB-XL. As before STEMI was compared with "Not STEMI with normal sinus rhythm". We simplified the table as follows and have added the results for LBBB:

Supplementary Table 3. Metrics of discriminatory performance in the external validation tests CODE-II (Brazilian) and PTB-XL (European). Only MI labels for the presence of STEMI (yes/no) are available in these external validation sets and can be evaluated **in addition to LBBB. In CODE-II, outcome-vs-rest comparisons are performed where the rest group consists of mostly normal ECGs. In PTB-XL, the outcome is compared with normal ECGs without any artifacts.** Arrows indicate the direction of better performance. The C-statistic and average precision (AP) are provided.

Metric	Outcome	CODE-II	PTB-XL
C-statistic (↑)	STEMI	0.9871	0.9977
	LBBB	0.9870	0.9979
AP (↑)	STEMI	0.5396	0.9936
	LBBB	0.8732	0.9988

3. Supplement, figure 1, panel B: change 80% in training to 70% to match what is found in the text

Thanks for spotting this typo! It should indeed be 70%.

The flowchart in SF1 needs further clarification. There are three main quantities: N ECGs > N patient visits > N patients. The sets (training, validation, test random, test temporal) are created so that a patient can only belong to one set (to prevent any information leakage between the sets). The validation and test sets are further restricted so they use at maximum one ECG per patient visit. ECGs from the same patient visit are expected to include highly correlated data and since some selected patients contribute with many ECGs this is not ideal for evaluation, where one unique record per patient visit is more suitable. On a patient level the split is 70% training / 10% validation / 20% test which the flowchart illustrates. However, exactly the same proportions are not seen for patient visits and ECGs (but similar), due to the mentioned restriction. However, we present the proportions on the patient level since it is on that level the split sets were created.

Updated figure:

We also clarified the description of the figure:

Panel A) Inclusion and exclusion criteria for the main study sample (the Swedish Emergency Department database [SwED]). There are three main quantities where N ECGs > N patient visits > N patients.

Panel B) Data splits of the derived main study sample. The sets (training, validation, test random, test temporal) are created so that a patient can only belong to one set (to prevent any information leakage between the sets). The validation and test sets are further restricted so they use at maximum one ECG per patient visit. The training, validation and random test set includes patients drawn at random whereas the temporal test set includes patients with their first available ECG recording during the last year (2016).

4. Supplement, figure 6: do you have a hypothesis for large difference in test performance for OMI, NSTEMI, LCX group? Consider mentioning the hypothesis as something to follow up on for future research. Could it be driven by a change in clinical practice over time to send more LCX STEMI equivalent to cath lab?

First we need to highlight the uncertainty of some performance metrics, especially for rare outcomes. The current "error bars" are as we write "minimum-maximum over ten trained models initiated with different seeds". This tells us something about how much the prediction performance differs depending on the random initiation of the model and the resulting local/global minima of the final epoch, i.e. different "optimal" solutions that the model may arrive at depending on initiation. It does however not give the full picture of the uncertainty of the

evaluation metric. The uncertainty by the test set sample might be more important, especially for rare outcomes. We have now added 95% confidence intervals for the evaluation metrics from a non-parametric bootstrap. There is a higher uncertainty for the smallest outcome categories, including LCx, so some minor differences should be interpreted with care as they could be explained by chance alone. However, with the 95% confidence interval added, the temporal test set still stands out with a better performance and lower uncertainty. We are not aware of any change in clinical practice that could explain this. Please also see responses to subsequent comments about the anatomical reasons for difficulties in LCX/RCA culprit differentiations.

Supplementary Figure 6. Discriminative performance (C-statistic) when comparing a given class (x-axis) with all other classes in the random or temporal test set **or combined**. All predicted sub classes of the model are included **as well as the super classes. 95% confidence intervals calculated from 2000 bootstrap draws.** The dashed horizontal line represents a random guess.

In the temporal test set we include all patients who had their *first* available ED visit, with an ECG taken, 2016. This to ensure that there is no patient overlap between the test sets and so that the temporal test set only includes patients and visits from the last year of follow up. However, this means that the random test set is more likely to include patients with multiple ED visits over multiple years (2005-2015). This does in effect include more elderly individuals and patients with prior CVD (who at average have more ED visits). The controls in the random test set are almost 10 years older than in the temporal and have a much higher frequency of prevalent CVD (sometimes threefold). Restricting the test sets to patients free from historic CVD has however not a huge impact on the temporal vs random difference. In summary, we don't have a good explanation for this and have added:

Compared with the temporal test set, the random test set included more patients with multiple ED visits in the SwED data (over a longer time span), were on average older, and had more prevalent cardiovascular comorbidities.

5. Supplement, figure 9: is binned calibration plot considered common knowledge? If not, please include a referens

Binned calibration curves and calibration evaluation metrics like the expected calibration error (ECE; a binned metric compatible with the binned calibration curves) are commonly used in machine learning (<https://proceedings.mlr.press/v70/quo17a/quo17a.pdf>). However, guidelines such as TRIPOD-AI suggest smoothed flexible calibration curves, i.e. continuous calibration instead of binned. TRIPOD-AI compliance is suggested by the journal so we have now added such calibration curves instead. The logistic calibration curves indicate acceptable calibration. In addition, smooth calibration curves fitted with a general additive model using thin-plate splines captures any non-linear trends together with the uncertainty. Some local trends are captured that deviates from the identity line, but with wide confidence bands in such areas and potentially driven by a few influential observations. We created these calibration curves for the larger superclasses as these have a higher number of cases and enough resolution in the calibration (as shown in the binned figures, many of the smallest subclasses had poor resolution/few bins due to few cases). We updated the methods description and figure:

We visualized the calibration of our model using **continuous** calibration plots of the observed case frequency vs the average predicted probability. We used the combined random and temporal test set for these plots to increase the total number of cases and **calibration resolution**. **Model calibration was assessed using both a logistic calibration intercept and slope and a flexible smooth calibration curve estimated via a penalized thin-plate regression spline within a generalized additive model.**

Supplementary Figure 9. Continuous calibration plots in both SwED tests combined. **Logistic calibration intercept and slope and a flexible smooth calibration curve estimated via a penalized thin-plate regression spline within a generalized additive model is presented. 95% confidence bands calculated from 2000 bootstrap draws.**

6. “Of these ECGs, 1,583 (0.3%), 4,279 (0.8%), and 534,510 (98.9%) cases were labeled at time of coronary catheterization by the coronary angiographer as OMI, nOMI (myocardial infarction not classified as OMI), and control (no myocardial infarction), respectively.” This sentence could be misinterpreted to mean that 500,000 controls were labeled as part of coronary catheterization.

We agree, and rephased this sentence:

Of these ECGs, 1,583 (0.3%), 4,279 (0.8%), and 534,510 (98.9%) were labeled as OMI, nOMI (myocardial infarction not classified as OMI), and control (no myocardial infarction), respectively. The OMI and nOMI labels were set at time of coronary catheterization by the coronary angiographer.

7. “...newly formed coronary total occlusion (TIMI blood flow 0).” Is there any guidance, explanation or reference that can be supplied to explain how “newly formed” is determined?

The distinction between an acute and a chronic occlusion can be made with certainty during the coronary intervention, based on the characteristics of the occlusion upon guidewire maneuvering. The definition used in this study is based on variables in the SCAAR registry indicating an occlusion of less than 3 months’ duration; the choice of 3 months for delimiting acute from chronic occlusions is somewhat arbitrary, but >3 months is the international consensus definition of a “chronic total occlusion”. In the experience of the consulted cardiologists responsible for setting up the registry and using these variables throughout its existence, occlusions classified as less than 3 months duration are typically new occlusions, as determined by their characteristics upon guidewire maneuvering. The distinction between an acute and a chronic occlusion is hence straightforward, but the exact age of the acute

occlusions can't be done just by the guidewire feel. The coding of these variables is done at the time of the coronary intervention with knowledge of the total patient presentation, including the timing of the onset of pain etc. We have clarified how newly formed vs chronic occlusion is defined:

OMI was defined as the presence of a newly formed coronary total occlusion or very close to total occlusion (TIMI blood flow 0). **The distinction between an acute and a chronic occlusion can be made with certainty during the coronary intervention, based on the characteristics of the occlusion upon guidewire maneuvering.**

8. Severe imbalance (OMI only 0.3% of ECGs) likely affects rare subclass performance, e.g., NSTEMI-OMI-LCX has very low AP despite good C-statistics. What specific measures were taken to account for this?

The low sample size for some specific outcomes is a limitation that we have now highlighted further in the discussion.

The number of cases in some of the outcome classes is modest (Supplementary Table 1), with quite few ECGs to learn from in training and few ECGs in evaluation leading to some uncertainty. Still, the results across outcome classes are encouraging, but some outcome classes such as the LCX culprits would likely benefit from a larger sample size in a future similar model.

A weighted loss function could have been considered where more weight is placed on the errors of the minority class. This is especially true if the model ignores the minority class in favor of the majority class. The total amount of available information (sample size of each class) still introduces a clear limitation. The fact that the model achieves good discrimination (C-statistic) with poor AP indicates that it has learned a discriminatory pattern but is quite unsure about itself. Here more data might have helped as we have now highlighted. It should also be added that although STEMI/NSTEMI has a clear value for the evaluation and training of the model, the main outcome of interest is OMI regardless of the presence/absence of a ST-elevation. So the larger superclass OMI-LCX is of higher clinical interest.

9. As mentioned above, it is not clear how the random and temporal test sets were created.

Yes, this should be clarified and for the flowchart in Supplementary Figure 1, we now write:

Panel A) Inclusion and exclusion criteria for the main study sample (the Swedish Emergency Department database [SwED]). **There are three main quantities where N ECGs $>$ N patient visits $>$ N patients.**

Panel B) Data splits of the derived main study sample. **The sets (training, validation, test random, test temporal) are created so that a patient can only belong to one set (to prevent any information leakage between the sets). The validation and test sets are further restricted so they use at maximum one ECG per patient visit. The training, validation and random test set includes patients drawn at random whereas the temporal test set includes patients with their first available ECG recording during the last year (2016).**

We hope that this clarifies how the test sets were created. In the main text we write:

We split the SwED data on patients (not ECGs), using 70% for training, 10% for validation, and 20% for testing. The test data comprised one random set and one temporal set including patients whose first visit occurred in the final study year, which is further described in Supplementary Figure 1.

10. The term “prevalent LBBB” is used several times. What does prevalent mean in this context?

We consider LBBB to be a chronic condition once it is present, so a patient is considered to have LBBB if the patient has any diagnosis of LBBB up until the time of the ED visit; this is what “prevalent” refers to. We also complement the diagnosis data with high probability predictions from an external, machine learning model using the ECGs at time of the ED visit. We thank the reviewer for noting that “prevalent” was not defined in the text. We extended the text (red) to make this clear.

LBBB classification was based on the LBBB predictions from an external, machine learning model trained on Brazilian data, which has been shown to outperform cardiology resident medical doctors in the classification of LBBB from ECGs. The model predictions were supplemented with prevalent ICD10:I44.6-I44.7 diagnoses from the patient registry and SWEDHEART annotations of LBBB, i.e. any diagnosis set up until the ED visit from all available diagnosis data of the patient. A patient was classified as having an LBBB at the ED visit if all ECGs of the ED visit had a $Pr(LBBB) > 0.5$ (cutoff selected by visual inspection of Supplementary Figure 11) or if a prevalent LBBB diagnosis was present in any of the other data sources.

11. Supplement, table 4: What is the reader expected to do with table 4? Are we expected to know the codes or are we expected to look up the codes or are we just expected to see repeated codes? I don't see how table 4 helps the reader understand the algorithm mistakes without looking up every listed code.

We fully agree with the reviewer that ICD10 codes alone are not enough when presenting these results. We have now added a description together with each code in an updated version ST4. A snapshot of what the new table looks like is provided below:

Truth	Misclassification	Over/under represented diagnosis	OR
Control	nOMI	Intentional self-harm by hanging, strangulation and suffocation (X70)	335.0
Control	nOMI	Asphyxiation (T71)	297.2

12. “Half of the false negatives showed clear STEMI patterns,...” This would be troubling to a clinical user. Is there any explanation that could help give the clinical user confidence that clear STEMI will not be missed?

We agree that the model should have few false negatives for clear STEMI cases, but this can at least partly be explained by the challenges in the CODE-II ECG data described below. The false negatives were not obvious STEMI cases after an additional review of the ECGs by a cardiologist. They were cases with ST elevation associated with other ECG abnormalities, such as left intraventricular conduction disturbances and atrial fibrillation with a high ventricular rate. These ECG alterations make the evaluation of the ST segment challenging, including for a physician. It is also important to clarify that the CODE-II database lacks paired coronary angiography, so we cannot confirm that these are acute coronary occlusions. In fact, this is a limitation of CODE-II. To summarize, the false negative STEMIs appear to be a mixture of label noise and challenging ECGs with other abnormalities which makes the interpretation challenging, even for a human. Especially since no confirmatory angiography report is available.

In the results we have added:

The false negatives with a clear ST elevation had other ECG abnormalities, such as left intraventricular conduction disturbances and atrial fibrillation with a high ventricular rate.

As part of the limitation of the discussion we have added.

The CODE-II dataset lacks paired coronary angiography, so it should be noted that acute coronary occlusion cannot be confirmed for those with a STEMI label.

13. "...perimyocarditis still raises some confusion for the model..." why not make perimyocarditis another model output like LBBB?

We also discuss this in general comment #1 of reviewer #1. Probability of perimyocarditis is an output of the model. It is however assumed to be mutually exclusive with MI since the two diagnoses almost never co-occur. This is to handle the main issue: controls, free from MI, with perimyocarditis, where perimyocarditis is mistaken for STEMI due to a STE-like pattern. The model has some discriminatory performance of perimyocarditis but the mixup with STEMI still occurs, probably because perimyocarditis and STEMI simply have ECG patterns that look very similar, hence still prone to misclassifications.

14. Conclusion: "...guide the angiographer to the critical coronary branch, in the case of concurrent chronic occlusions" Is the precision high enough for the cath lab user to confidently stent the predicted culprit first without a complete survey of coronary artery tree? This conclusion seems a bit strong.

A complete coronary angiography is always performed, before any treatment of the findings ensues. We realise that our previous wording may come across as strong. We have now changed it to "... inform the angiographer on which is the most likely culprit, ...".

15. "OMI was defined based on the SWEDEHEART SCAAR variable OCKL. The culprit vessel of the OMI was defined as the segment marked as having the newly formed occlusion." Is the newly formed occlusion a very confident diagnosis? If there is doubt, it should be mentioned in the study limitations.

We agree that this section needs to be clarified; please see comment 7 above.

16. Outcome combinations: Is it correct that the MI label means acute rather than old?

MI (OMI+nOMI) means a new onset MI (ICD10:I21-I23) with hospitalization within 24h after the ED visit. Exactly when the patient got their MI is unknown, we only know when they arrived at the ED with their presented complaint. Most patients are expected to seek healthcare soon after symptom onset, but some rare cases have waited for days before seeking healthcare for a MI. New onset MI is acute, but the OMI subset is extra acute and requires urgent intervention as highlighted in the paper. The definition of MI (ICD10:I21-I23) and OMI based on the SWEDEHEART SCAAR OCKL variable has been clarified as mentioned in the previous comments. A patient with an historic MI hospitalization is treated as control, given that it took place >30 days before the control's ED visit date. MIs in the time period 30 to 1 days before the ED visit date are excluded as described in the previous paper. We have added:

MI was defined as a new onset acute myocardial infarction (ICD10:I21-I23 set as diagnosis) as described previously,¹² and the sub-class OMI was defined as a newly formed occlusion with complete or close to complete blockage requiring more urgent intervention ...

17. Data splitting: Just a comment – it would be interesting to see the performance if a human chose the best ECG for a patient rather than a random ECG. This would mimic the paramedics when they have the right constellation of symptoms.

Here it should be noted that in training we use all available ECGs of a patient visit. Even if one specific ECG is the most informative for a human, we hope to also detect subtle non-human detectable patterns with the machine learning model, so training on all ECGs connected to an ED visit makes sense from that perspective. It also makes sense as a form of data augmentation to use multiple ECGs compatible with the end diagnosis of the ED visit. In the test sets, however, we select one ECG per patient visit at random. Selecting these ECGs manually

would be too time-consuming and it would only work for already human-detectable patterns. The current approach should be conservative by sometimes including ECGs with a less obvious pattern in the evaluation (when multiple ECGs are available). It should also be noted that the 10 second ECG snapshots of the hospital ECG database are not random but rather a segment of the ECG that the health care staff has decided to save in the database, i.e. manually saved for a reason. It is the selection of one ECG per patient visit in the test sets that is done at random when multiple ECGs are available for a patient visit.

18. Model validation sets: It is not clear what “manually curated” means for PTB-XL ECG choice. Does it mean that there were more than 75 acute MI cases but 75 were chosen?

We did a manual review of the PTB-XL records to confirm the presence/absence of a ST-elevation to validate the PTB-XL labels. This was done by a senior cardiologist (JS) as we describe below. In the original paper that we refer to we included 75 MIs with a **probable or definite** ST-elevation. In this paper we focus on the 63 MIs that had a definite ST-elevation based on the review of the senior cardiologist. The main purpose of the PTB-XL external validation set in this work was to evaluate how the model performed for noise-free ECGs with a clear ECG human-readable pattern. The CODE-II dataset and newly added InCor external validation set include more challenging ECGs. We have added some further explanation in the **Supplement**. In the main text we write:

From PTB-XL, 63 acute myocardial infarction cases with a confirmed ST-elevation, 279 LBBBs, and 197 randomly selected **normal** controls without ST-elevation were manually curated from the PTB-XL database. The inclusion criteria of the PTB-XL external validation set used in this study has been described previously¹² and is also summarized in the **Supplement**.

In the Supplement we write:

PTB-XL external validation set

The PTB-XL is a publicly available database of 21,837 10-second 12-lead ECGs annotated with 71 different ECG statements.^{13, 14} From PTB-XL we included ECGs that had been evaluated by at least one PTB-XL cardiologist (validated_by_human=True) and with no electrode problems reported (empty electrodes_problem). This resulted in 16,065 ECGs, and from these we (before running any models) extracted:

- 63 STEMI ECGs. In total, 98 ECGs are annotated as likely acute myocardial infarctions (infarction_stadium1="Stadium I" or infarction_stadium2="Stadium I"). A senior cardiologist (JS) reviewed the ECGs for the presence of a ST-elevation and 63 ECGs with a definite ST elevation-pattern were kept.
- 279 LBBB ECGs. All ECGs labeled as complete LBBB (CLBBB:100) with available age and sex information were kept.
- 197 normal ECGs. In total, 200 likely normal controls without myocardial infarction were randomly drawn (empty infarction_stadium1 and empty infarction_stadium2 and NORM:100). Only a random subset of 200 was drawn to be able to complete a manual review within reasonable time. A senior cardiologist (JS) reviewed the ECGs for the presence of a ST-elevation and 197 ECGs without a definite ST elevation-pattern were kept.

Age and sex was extracted from the PTB-XL database. Age was missing for one STEMI ECG and imputed with the mean age of STEMIs in PTB-XL. Age was further normalized using the mean and standard deviation from the training dataset.

19. Model performance in subgroups: it would be helpful to indicate which tables and figures represent the text. The reader can go back and forth to determine which figure/table but I suggest adding references to make it easier for the reader.

This is a good point to refer the reader to the corresponding table/figure so we have added:

Certain subgroups remain an issue; perimyocarditis still raises some confusion for the model, and discrimination of nOMI in the LBBB subgroup is poor but better than chance (Supplementary Table 4, Supplementary Figure 10).

20. No detailed breakdown of performance by sex, ethnicity, or comorbidity burden in external data.

This is a good suggestion and we have now added stratified evaluation metrics for CODE-II. CODE-II, unlike the two other small external validation sets, allows for granular stratification. We write:

The performance was mostly comparable across age, sex, and comorbidity strata in the CODE-II dataset (Supplementary Table 12-13)

Supplementary Table 12. Stratified metrics of discriminatory performance in the external validation test CODE-II (Brazilian). The outcome groups STEMI and LBBB are tested versus the rest group consisting of mostly normal ECGs. Strata include sex, age group (years), and comorbidity burden (defined as the presence of zero or at least one of seven pre-specified clinical conditions). Arrows indicate the direction of better performance. The C-statistic (C) and average precision (AP) are provided.

		Sex		Age group (years)						Comorbidities	
	Outcome	Female	Male	18-39	40-49	50-59	60-69	70-79	≥80	0	≥1
C (↑)	STEMI	0.9899	0.9827	0.9721	0.9880	0.9943	0.9887	0.9773	0.9792	0.9896	0.9851
C (↑)	LBBB	0.9943	0.9743	0.9368	0.9717	0.9829	0.9866	0.9869	0.9824	0.9837	0.9866
AP (↑)	STEMI	0.5656	0.5297	0.3062	0.5244	0.5582	0.5828	0.5616	0.5319	0.5379	0.5423
AP (↑)	LBBB	0.9205	0.8066	0.5414	0.7910	0.8587	0.8860	0.8952	0.8870	0.8482	0.8801

21. No use of saliency maps, Grad-CAM, or lead-level attribution to understand model focus. Clinicians will require interpretability to build trust and integrate into workflows—especially for localization claims.

This is a good suggestion and we have now performed a Grad-CAM analysis. However, the results proved challenging to interpret as the final convolutional layer yielded diffuse activations that highlighted large parts of the entire ECG signal uniformly, regardless of the class predicted. This lack of localization persisted across multiple post-processing attempts, including power transformations and scaling approaches. Interestingly, more localized and human-readable features were observed in earlier blocks, yet the "optimal" layer for visualization shifted inconsistently depending on the target outcome. Example RCA STEMI:

The Grad-CAM results were deemed uninformative for conclusive interpretation and were subsequently excluded from the final analysis.

22. Manual re-review (Table 5) reveals that many “false positives” were actually STEMI cases upon expert review, suggesting label noise in the reference standard—this may underestimate true performance.

We appreciate the reviewer's thoughtful observation regarding label noise in the reference standard. However, the inclusion of expert adjudication actually strengthens rather than undermines our findings, as it demonstrates the model's ability to identify ECGs that are diagnostically challenging and benefit from expert review. We acknowledge that the true sensitivity may be slightly higher if all of these cases were definitively confirmed by coronary angiography. However, this is not available in CODE-II and represents a limitation that we have now highlighted:

The CODE-II dataset lacks paired coronary angiography, so it should be noted that acute coronary occlusion cannot be confirmed for those with a STEMI label.

23. Performance across device types (MAC55 vs others) and software versions varied modestly, suggesting potential site- or vendor-specific tuning effects. This should be noted.

There is indeed some modest performance metric variability depending on technical factors. First, it should be noted that these are univariable stratified analyses. The difference between device types could be driven by the fact that one device type is used at e.g. a care unit with more patients with LBBB, which might then be the true driving factor between device differences (that is unknown in the current analysis). Secondly, these performance metrics come with an uncertainty related to the test set sample. If we calculate 95% confidence intervals around the performance metric by non-parametric bootstrapping we get the following figure which indicates that some of the very modest differences might be driven by random differences in the test set samples. So in summary, small differences are seen between strata that might be explained by chance alone, with the exception of some larger differences seen primarily for age, previous CVD, and prevalent LBBB. The difference between age groups could also be driven by a large difference in LBBB and CVD prevalence between age groups.

Reviewer #2 (Remarks to the Author):

Reviewer #3 (Remarks to the Author):

This studies extends the DNN models developed for prediction/diagnosis of myocardial infarction in emergency department patients (PMID: 36380048). In a large number of ECGs (N=540,372) from the Swedish Emergency Department Database paired with catheterization outcomes a deep learning model was trained and validated for the diagnosis of (non-) occlusion myocardial infarctions reaching a C-statistic of >0.95 for occlusion myocardial infarction, >0.87 for non-occlusion myocardial infarction.

High constant performance across subgroups was observed, in particular different ECG hardware/software. Even on left bundle branch block ECGs the model performed well.

Specific comments:

-More emphasis could be place on nOMI, a more common condition and often more difficult to decide on a culprit lesion.

This is a very interesting suggestion for a future project. It however introduces challenges due to the data we have available to us. For OMI, typically a single vessel is marked as the new occlusion by the angiographer. For nOMI, no such label is available for the culprit in our registry data. Also, in the absence of an occlusion, it may be difficult to know which of the high-grade stenoses (in case of multiple coronary lesions, which are common) are in fact causing the infarction. It could potentially be defined based on the percent blockage of the vessel, but old occlusions with collaterals formed can have a high %occlusion but still don't be the culprit. Alternatively, the vessel(s) where a stent was inserted could be considered, but that might still not identify the main culprit, and with noise introduced by balloon treatments without stents, failed stenting attempts, and multiple stent placements (with amount of stenting varying over time, and the optimal amount of stenting currently being a debated topic). We will consider this for future development but with the current data we have it is not obvious how to set this up, and we propose keeping OMI as the main focus of this analysis.

-How much better does the current model perform compared to existing, automated STEMI diagnosis by the ECG machine algorithms?

Several proprietary algorithms for automatic ECG diagnosis statements exist on the market, from different ECG machine vendors, often requiring a license to run. We don't have access to most such licences so we focus our evaluation of the rule-based ECG statements that come with GE MUSE as default - Marquette 12SL. We extracted all statements from the GE MUSE XML files in our data export and extracted the statements for STEMI. We write in the Supplement:

12SL diagnosis statements

From each GE MUSE ECG of the SwED database, we extracted all automatic ECG diagnosis statements from Marquette 12SL. An ECG with the following statements (in Swedish) were labeled as STEMI: "*** ** AKUT MYOKARDINFARKT ** **" and "*** ** AKUT MI/STEMI ** **". The model of the present study outputs predicted probabilities. When thresholding the probabilities to "call" a given outcome, many aspects should be considered, including disease prevalence, risk vs benefit, health economics, healthcare load, and ethics. No such analysis has been performed. In this analysis we set a threshold that resulted in a comparable number of false positives to the 12SL statements.

We add:

We also compared the model's prediction of STEMI versus the 12SL automatic diagnosis statements as described in the Supplement.

We summarize the results as:

In both SwED test sets combined, the model captured significantly more STEMI compared with the automatic 12SL diagnosis statements (part of the GE MUSE data format) at the same number of false positives (Supplementary Table 10-11). A total of six STEMI ECGs, that were correctly called in the diagnosis statements, had a low predicted probability of STEMI from our model. A manual evaluation of these by a cardiologist (JS), revealed that all except one had an unclear ST-pattern that should not result in a confident STEMI diagnosis. When it comes to missed STEMI, it should be noted that our dataset contains some STEMI label noise as shown in a previous study.¹² The STEMI label was determined at discharge from the coronary care unit or emergency room when the whole care episode could be summarized. The ECGs in the test sets of this study may hence not always be the ones guiding the final diagnosis.

Supplementary Table 10. STEMI vs prediction of STEMI, from 12SL (automatic diagnosis statements in the GE MUSE format) and the model of the present study. From 12SL no predicted probabilities are available, only called outcome labels. For the present study's model a probability cutoff of STEMI is set so that the proportion of false positives is comparable to 12SL. Results of truth vs prediction are presented for both SwED test sets combined. *6 ECGs out of 71 misclassified ECGs from the study model were correctly classified by 12SL; a manual review of those ECGs by a senior cardiologist (JS) revealed cases with unusual patterns, high heart rates, disturbances, and possible label noise (e.g. early depolarisation patterns), that could all be possible explanations for the confusion.

	Truth	Prediction	Number
12SL	No STEMI	No STEMI	75503
	STEMI	STEMI	133
	No STEMI	STEMI	402
	STEMI	No STEMI	138
Study model	No STEMI	No STEMI	75511
	STEMI	STEMI	200
	No STEMI	STEMI	394
	STEMI	No STEMI	71*

Supplementary Table 11. Rows represent STEMI vs prediction of STEMI, from 12SL among all with STEMI-OMI. *Pr(STEMI) is the median (IQR) predicted probability of STEMI from the study's model. **%STEMI is the proportion of ECGs predicted as STEMI by the study's model.

Truth	Prediction	N	Pr(STEMI), model*	%STEMI, model**
STEMI-OMI	STEMI	98	0.96 (0.76,0.99)	95%
STEMI-OMI	No STEMI	71	0.16 (0.01,0.67)	59%

-The authors select important potential confounders. Should medication intake also be considered as a confounding variable (e.g. digitalis)?

This is a very good suggestion, but unfortunately we do not have enough data to answer this specific point. If we look at the number of ECGs where a patient had ≥ 1 dispensation of digoxin (ATC:C01AA05) in the entire 90 day period prior to the ED admission, we only have 1,256 records in total (out of these, 5 are nOMI and 0 OMI). If we instead look at a diagnosis of

drug intoxication known to primarily affect the heart (ICD10:T46), which includes heart stimulating glycosides, we only have 9 records at time of the ED visit in total. In both cases, we do not have enough records to set up a reasonable analysis.

In Supplementary Table 4, we analysed diagnoses that were over/under-represented among high-probability misclassifications of the model. We now performed a corresponding analysis for medication from the Swedish prescribed drug registry. All 5-character and 7-character ATC codes of drugs (corresponding to a category of drugs or a given compound) dispensed in the 90 day period prior to the ED visit were considered. Some of the top results, significant after correction for multiple testing, included drugs with a known side-effect on the ECG but many had a more unclear connection to the ECG. Some medications might just co-vary with an underlying disease that in turn affects the ECG. These were the top findings:

moxifloxacin, drugs related to hyperkalemia/hyperphosphatemia, hypothalamic hormones, gonadotropins, insulin pump with infusion set, analgesics including codeine and ibuprofen, anti-pulsives (loperamid), anti-anemics, statins, nitrates, ASA, ACE/ARB.

-The worse performance for LCX/RCA localization is not specific to the model, but also seen for human diagnosis. It is based on anatomical reasons because the ECG mirrors the occlusion location which can be a large RCA or an LCX depending on the vessel size and epicardial course of the artery. The authors may want to comment on this and the ability of future and different models to overcome this limitation.

The reviewer highlights an important challenge and limitation that is driven by anatomical reasons but potentially also with a contribution of small sample sizes for some outcome classes. We agree that this should be discussed further and have added:

Certain subgroups remain an issue; perimyocarditis still¹² raises some confusion for the model, and discrimination of nOMI in the LBBB subgroup is poor but better than chance. **The challenge in separating between OMIs with occlusions in LCX and RCA is not unexpected; the origin of the posterior descending artery is more often the RCA than the LCX (Fig 1. Panel A+B). This discrimination is challenging also for humans using ECGs.[PMID: 19706635] The number of cases in some of the outcome classes is modest (Supplementary Table 1), with quite few ECGs to learn from in training and few ECGs in evaluation leading to some uncertainty. Still, the results across outcome classes are encouraging, but some outcome classes such as the LCX culprits would likely benefit from a larger sample size in a future similar model.**

-Were posterior leads (v7-9) available that could have led the model to identify LCX OMI more easily?

This is an interesting suggestion but unfortunately, no, we only have 12-lead data with some very rare exceptions. Since our model does not allow a variable lead configuration we cannot include ECG leads that are only available in a subset of the patients.

-The findings should be set into the context of recent studies such as Cho KH Heart J Digit Health. 2025, PMID: 40703113, Büscher A Eur Heart J. 2025 PMID: 40156923.

The suggested articles are indeed interesting and related to the current study. Their definition is however based on who received revascularization according to current care. With our definition we hope to capture patients with an urgent need of vascularization regardless of how the patient was handled in current care. Our OMI label, set at time of catheterization, includes all acute occlusions irrespectively of treatment result. Specifically, the article by Cho et al presents a great technological development of the field by using a transformer-based model, but their outcome was acute myocardial infarction, not OMI, and STEMI/NSTEMI were also not used as outcomes but only as analysis strata.

Buscher et al also studied an outcome that isn't really what we mean with OMI (was revascularization performed (yes/no) and a definition based on troponin). As he writes: *Coronary revascularization at the time of patient presentation was chosen as a pragmatic endpoint for model training due to limited clinical details provided with the MIMIC-IV dataset and the large sample size prohibiting individual, manual outcome adjudication.*

-Is there a potential for bias if patients who did not undergo catheterization could have missed as nOMI or, less likely, OMI?

There are indeed a few important such situations to consider. The most important scenario is the following: All certified first responders (paramedics) in Sweden have access to prehospital ECG and telemedicine tools. When indicated, they obtain a standard 12-lead ECG in the field and send it to a hospital cardiologist that makes a decision based on that ECG to activate or not activate the cath lab; if the cath lab is activated, the patient is directly sent to the cath lab, typically bypassing the ED and without leaving traces there in the form of troponins or ED ECGs. So these STEMI OMI did get an angiography but are absent from our database because they did not get an ED registration or ED ECG. Hence our ED ECGs on average represent slightly more challenging cases than these textbook STEMI examples, that are either walk-ins or more complex presentations. Another scenario to consider is frail patients that do not get an angiography because it is considered a risk greater than the benefit. These will lack angiography results and hence do not have the labels used in this study. In a recent trial in elderly patients with NSTEMI, an invasive strategy did not improve outcomes: <https://www.nejm.org/doi/full/10.1056/NEJMoa2407791>. So the frail group may benefit less from these model predictions. Nonetheless, there is a trend for more elderly NSTEMI patients to be treated invasively over time (<https://www.ucr.uu.se/swedeheart/dokument-sh/arsrapporter-sh/01-swedeheart-annual-report-2024-english-2/viewdocument/3703>).

Another source of missingness is if the MIs were missed altogether, of course.

To conclude, would this missingness cause bias? If all consecutive patients would be included, the results of these scenarios would be true myocardial infarctions among the so-labelled non-cases, hence label noise. This would tend to bias the models towards poorer performance. One way that we have mitigated this potential label noise is that among the non-cases (not MI) we require that they do not have an MI diagnosis within [-30, 7] days around the ED visit, so controls should not be contaminated with missed infarctions, if they showed up in healthcare. Sweden has complete coverage of all diagnoses in the national registries, so only the scenario of completely missed MIs would still be a problem. We nevertheless believe that our current sample is representative of a future target population that might use the model - as long as an ECG was obtained for the patient.

-Whereas the baseline characteristics of the cohort are important they could be moved to the supplement if this would allow for more data on the central results in the main manuscript.

Yes, we agree. Population characteristics should be included for the reader's understanding of the cohort, as suggested by reporting guidance checklists like TRIPOD-AI. These characteristics can however be briefly summarized in the main text and then presented in a supplementary table. **Table 1 has now been moved to Supplementary Table 6.**

Currently, the only important results presentation in the context of this study is Figure 1, panel C.

-The legend of the figure could explain nOMI vs. NSTEMI, OMI vs. STEMI.

We agree that panel C in Figure 1 is the most important part as it shows key evaluation metrics of the model. We think the other panels are useful for readers not familiar with the coronary artery tree segments and the outcomes studied. Figure 1 should be more standalone from the rest of the text and fully describe the key outcomes as the reviewer correctly points out. We have added in red:

... Panel C shows the discriminative performance (C-statistic) when comparing a given class (x-axis) with all other classes in the random or temporal test set. The dashed horizontal line represents a random guess. Different sub-classes of myocardial infarction (MI) are evaluated: with/without occlusive MI (OMI/nOMI), with/without the presence of a ST-elevation (STEMI/NSTEMI), and OMI culprit vessel localization (left main or left anterior descending artery [LM/LAD], the left circumflex artery [LCX], or the right coronary artery [RCA]).

-As the authors state, the value of the model is in the early/preclinical phase to identify patients with high risk of OMI. In this setting the exact localization of the occluded artery probably is not of high importance. They might explain in more detail what the benefit of knowing the occlusion site in OMI in the cathlab is? Both vessels are examined routinely.

We agree that OMI vs not OMI provides most of the clinical value. Knowing where the occlusion is located directs the PCI operator to the right place more quickly but is of more limited clinical value, and mostly in the case of multiple occlusions of which one is acute. A complete coronary angiography is always performed, before any treatment of the findings ensues. We realise that our previous wording may come across as strong. We have now changed it to "... inform the angiographer on which is the most likely culprit, ...".

-In clinic, lead III plays an important role for the diagnosis of STEMI, would this lead not add additional information?

The reviewer is completely right in that lead III can be visually useful for a human examiner. However, lead III can be fully expressed as a function of two other leads ($III = II - I$, from Einthoven's law), so the neural network will get complete information about III from I+II. I, II, V1, V2, V3, V4, V5, V6 are in fact the only leads that are stored in many ECG formats (like GE MUSE), because the remaining leads in a 12-lead setup can be directly calculated from these 8 leads.

-Since it is central for the understanding, please explain the definition of OMI (SWEDEHEART SCAAR variable OCKL) and STEMI in more detail. Does "old chronic occlusions" refer to "chronic total occlusions"? How many old chronic occlusions were observed? Is a differentiation for nOMI into type 1 and type 2 myocardial infarction possible?

Please see the response to comment #7. The coding of these variables is done at the time of the coronary intervention with knowledge of the total patient presentation, including the timing of the onset of pain etc. In order to clarify we have added this update:

The OMI and nOMI labels were set at time of coronary catheterization by the coronary angiographer. OMI was defined as the presence of a newly formed coronary total occlusion or very close to total occlusion (TIMI blood flow 0). The distinction between an acute and a chronic occlusion can be made with certainty during the coronary intervention, based on the characteristics of the occlusion upon guidewire maneuvering.

...

MI was defined as a new onset acute myocardial infarction (ICD10:I21-I23 set as diagnosis) as described previously,¹² and the sub-class OMI was defined as a newly formed occlusion with complete or close to complete blockage requiring more urgent intervention as opposed to chronic occlusions/stenoses in the other MI cases that have slowly been built up (where the heart, to some extent, might have adapted, by forming collateral vessels bypassing the blockage).

We don't have information on the number of chronic total occlusions. The data do not allow for classification into type 1 and 2 infarctions; variables to capture that have now been added to the registry but they were not in the registry during the observation period.

-In the baseline table it might be of interest to know how many stents were placed.

This is a good suggestion to highlight if there is significant difference between OMI vs nOMI with regard to treatment. We have added in the text:

The OMI cases also had markedly elevated levels of cardiac troponin levels, an elevated risk of death within 30 days (Supplementary Table 6), all underwent percutaneous coronary intervention (PCI) within the first week after ED admission (Supplementary Figure 5), **and among those who received stents, 36% required multiple stents (compared to 30% for nOMI).**